# `dopanim`: A Dataset of Doppelganger Animals with Noisy Annotations from Multiple Humans

**Marek Herde**[*]  **Denis Huseljic**  **Lukas Rauch**  **Bernhard Sick**

University of Kassel, Hesse, Germany
[*]`marek.herde@uni-kassel.de`

## Abstract

Human annotators typically provide annotated data for training machine learning models, such as neural networks. Yet, human annotations are subject to noise, impairing generalization performances. Methodological research on approaches counteracting noisy annotations requires corresponding datasets for a meaningful empirical evaluation. Consequently, we introduce a novel benchmark dataset, `dopanim`, consisting of about 15,750 animal images of 15 classes with ground truth labels. For approximately 10,500 of these images, 20 humans provided over 52,000 annotations with an accuracy of circa 67%. Its key attributes include (1) the challenging task of classifying `doppelganger animals`, (2) human-estimated likelihoods as annotations, and (3) annotator metadata. We benchmark well-known multi-annotator learning approaches using seven variants of this dataset and outline further evaluation use cases such as learning beyond hard class labels and active learning. Our dataset and a comprehensive codebase are publicly available to emulate the data collection process and to reproduce all empirical results.

## 1 Introduction

Supervised learning with machine learning models, such as neural networks (NNs), requires annotated data. Typically, multiple human annotators, e.g., crowdworkers [59], are tasked to provide the corresponding annotations, e.g., class labels. These annotators perform differently for various reasons, including bias, fatigue, and ambiguity in interpretation [16]. As a result of such imperfect annotator performances, we obtain noisy annotations that can significantly degrade models' generalization performances [50]. Therefore, annotations are often requested from multiple annotators per data instance. The intuition is that the majority vote is a reliable estimate of the ground truth annotation ("wisdom of the crowd"). However, such an approach leads to substantially higher annotation costs and ignores the annotators' different performances. More advanced approaches have been proposed to improve NNs' robustness against noisy annotations. These advancements include new regularization techniques [72], loss functions [76], and approaches to estimate annotator performances [43]. The evaluation of such approaches is primarily driven by empirical research, which necessitates access to datasets that realistically reflect the noise induced by human annotators [62]. However, due to the often high annotation costs, the number of publicly available datasets for methodological research is relatively low. Moreover, the potential ability of certain humans to self-assess their own knowledge and uncertainty is typically not queried as part of the annotation campaign. Accordingly, most existing datasets are limited to classification tasks that require no expert knowledge, provide only hard class labels as annotations, and lack metadata [75] about the annotators.

Motivated by the critical impact of noisy annotations in practical applications and the scarcity of corresponding datasets, we publish a novel dataset with the following profile and contributions:

38th Conference on Neural Information Processing Systems (NeurIPS 2024) Track on Datasets and Benchmarks.

> **dopanim: Profile and Contributions**
>
> (1) The dopanim **dataset**[a] features about 15,750 animal images of 15 classes, organized into four groups of doppelganger animals and collected together with ground truth labels from iNaturalist [21]. For approximately 10,500 of these images, 20 humans provided over 52,000 annotations with an accuracy of circa 67%.
>
> (2) **Key attributes** (cf. Fig. 1) include the challenging task of classifying doppelganger animals, human-estimated likelihoods per image-annotator pair, and annotator metadata.
>
> (3) The dataset's **broad research scope** covers noisy label learning [50], multi-annotator learning [39], active learning [16], and learning beyond hard labels [69].
>
> (4) We **benchmark** multi-annotator learning approaches on seven variants (annotation subsets) of our dataset to evaluate their robustness against annotation noise in diverse scenarios.
>
> (5) Three illustrative evaluation **use cases** regarding human-estimated likelihoods [8], annotator metadata [75], and annotation times in active learning [42] outline our datasets' potential to explore further research fields of machine learning.
>
> (6) A comprehensive **codebase**[b] enables emulating our data collection, reproducing all our experiments with dopanim, and includes other related datasets for a unified evaluation.
>
> ---
>
> [a]https://doi.org/10.5281/zenodo.11479589
> [b]https://github.com/ies-research/multi-annotator-machine-learning/tree/dopanim

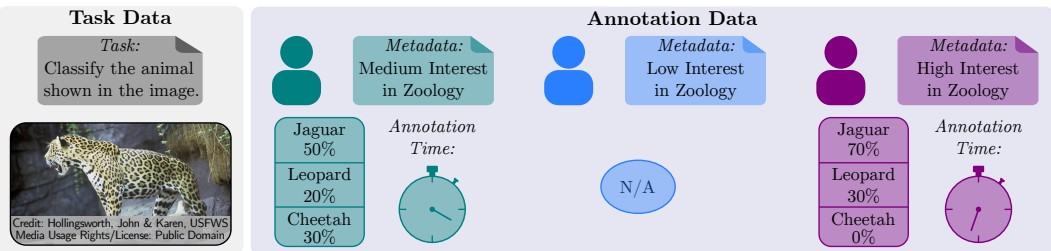

Figure 1: Simplified illustration of the data types included by dopanim – Two of three annotators provide probabilistic labels (after normalization) to identify the animal in the image. In addition to these annotations, annotation times and annotator metadata, e.g., interest in zoology, are available.

This article's remainder is structured as follows: Section 2 discusses related datasets. A description of the data collection process of our dataset dopanim and exemplary analyses are part of Section 3. We introduce variants of our dataset in Section 4 for benchmarking multi-annotator learning approaches. Section 5 presents further use cases of our dataset. We conclude our work in Section 6.

## 2 Related Datasets

Learning from noisy annotations covers diverse problem settings, which mainly differ in the learning tasks and their assumptions about the annotations' origin. For example, publicly available datasets with noisy annotations from humans exist for image segmentation [75] and sequence classification tasks [43]. Other datasets are scraped from the web [66, 29], where noisy annotations arise due to unreliable web resources. We focus on basic classification tasks with annotations from multiple humans for two reasons: (1) Classification tasks are the most common research topic in learning from noisy annotations and often serve as starting points for extensions to other learning tasks [50]. (2) Due to crowdsourcing and annotation companies, human annotators are a popular resource for annotating datasets [73]. Along with these annotations, we can easily get information on which annotation originates from which annotator. With this scope, we use Table 1 in the following to discuss popular and publicly available datasets regarding their task and annotation data, including dopanim for comparison. Further datasets with partial relevance to our scope are detailed in the appendices.

Table 1: Dataset overview – Column headings indicate the names of the datasets, whereas rows provide information regarding different attributes and statistics of the datasets. We denote counts by the # symbol and estimates by the ∼ symbol. Means are supplemented by standard deviations.

| Dataset Overview | spc [44] | mgc | labelme [43] | cifar10h [37] | cifar10n [62] | cifar100n | cifar10s [8] | animal10n [49] | dopanim ours |
|---|---|---|---|---|---|---|---|---|---|
| Task Data | | | | | | | | | |
| data modality | text | audio | image | image | image | image | image | image | image |
| training instances [#] | 4,999 | 700 | 1,000 | 10,000 | 50,000 | 50,000 | 1,000 | 50,000 | 10,484 |
| validation instances [#] | ✗ | ✗ | 500 | ✗ | ✗ | ✗ | ✗ | ✗ | 750 |
| test instances [#] | 5,428 | 300 | 1,188 | 50,000 | 10,000 | 10,000 | 50,000 | 5,000 | 4,500 |
| classes [#] | 2 | 10 | 8 | 10 | 10 | 100 | 10 | 10 | 15 |
| Annotation Data | | | | | | | | | |
| annotators [#] | 203 | 44 | 59 | 2,571 | 747 | 519 | 248 | 15 | 20 |
| annotation platform | AMT | AMT | AMT | AMT | AMT | AMT | Prolific | N/A | LabelStudio |
| annotator metadata | ✗ | ✗ | ✗ | ✗ | ✗ | ✗ | ✗ | ✗ | ✓ |
| annotation times | ✗ | ✗ | ✗ | ✓ | ✓ | ✓ | ✓ | ✗ | ✓ |
| soft class labels | ✗ | ✗ | ✗ | ✗ | ✗ | ✗ | ✓ | ✗ | ✓ |
| annotations per instance [#] | $5.6_{\pm 0.7}$ | $4.2_{\pm 2.0}$ | $2.5_{\pm 0.6}$ | $51.4_{\pm 1.5}$ | $3.0_{\pm 0.0}$ | $1.0_{\pm 0.0}$ | $6.2_{\pm 0.4}$ | $1.0_{\pm 0.0}$ | $5.0_{\pm 0.2}$ |
| annotations per annotator [#] | $137_{\pm 346}$ | $67_{\pm 106}$ | $43_{\pm 41}$ | $200_{\pm 0}$ | $201_{\pm 329}$ | $96_{\pm 233}$ | $25_{\pm 0}$ | $\sim 4{,}000_{\pm 0}$ | $2{,}602_{\pm 1{,}255}$ |
| overall accuracy [%] | 78.9 | 56.0 | 74.0 | 94.9 | 82.3 | 59.8 | 84.4 | $\sim 92.0$ | 67.3 |
| accuracy per annotator [%] | $77.1_{\pm 17.1}$ | $73.3_{\pm 24.4}$ | $69.2_{\pm 18.2}$ | $94.9_{\pm 0.1}$ | $82.1_{\pm 6.1}$ | $55.6_{\pm 27.3}$ | $84.4_{\pm 8.1}$ | N/A | $65.6_{\pm 14.7}$ |

## 2.1  Task Data

Under task data, we summarize the data essential for defining the classification task. Most datasets focus on image classification. In particular, the datasets providing noisy annotations for the popular image benchmark data `cifar10`, `cifar100`, and `labelme` are used for generic object classification. The dataset `mgc` deals with the classification of music audio files according to their genre, while `spc` deals with the classification of the polarity of sentences. Many of these classification tasks share the characteristic that no special domain knowledge is required for correct classification. In contrast, our dataset `dopanim` considers the challenging image classification of four groups with classes of highly similar animals. Thus, our dataset enables the investigation of learning scenarios where annotators have varying levels of domain knowledge. The `animal10n` dataset also targets classifying animals but is limited to pairs of mildly confusable animals. Except for `cifar100n`, the number of classes in the other datasets tends to be low. This also applies to `dopanim` due to the already high complexity of distinguishing the 15 animal classes. Furthermore, all datasets provide predefined splits into training and test data, but only `labelme` and `dopanim` provide extra validation data to ensure reproducibility when optimizing hyperparameters. After filtering invalid images, e.g., ones showing only animal bones, the splits of `dopanim` contain approximately the same number of images per class.

## 2.2  Annotation Data

Under annotation data, we summarize the data related to the annotation process. The number of annotators is relatively high for almost all datasets, as these annotators are recruited via large crowdsourcing platforms such as Amazon Mechanical Turk (AMT) [2] and Prolific [38]. However, since a limited annotation budget has to be distributed among many annotators, the number of annotations per annotator is rather low. Accordingly, analyzing each annotator's behavior in depth is difficult. In contrast, the annotations of `dopanim` originate from fewer annotators, each of whom has provided many annotations via LabelStudio [55]. The high standard deviation in the number of annotations per annotator is typical in practice [60]. The dataset `animal10n` also includes numerous annotations per annotator, though their average accuracy is estimated to be relatively high. Conversely, the partially low overall annotation accuracies for `dopanim` and the other datasets combined with the quite large differences between the annotators' individual accuracies demonstrate the need for multi-annotator learning techniques. Beyond hard class labels, more informative annotation types can also be requested. In particular, soft class labels capture the annotators' subjective uncertainties regarding their decisions. For `cifar10s`, the probabilities for the two most probable class labels and any class label to which the image does definitely not belong are available. Our dataset `dopanim` provides soft class labels, where the annotators could distribute unnormalized likelihood scores across all class labels to reflect their uncertainties. Other important data can be collected in addition to the annotations when annotating. This includes metadata about the annotators [75], such as self-assessed motivation, and their annotation times. While annotation times are provided by some other datasets, detailed annotator metadata is only provided by `dopanim`.

# 3   `dopanim`: A Dataset of Doppelganger Animals

This section describes the task and annotation data collection. Further details, including ethical considerations, are given in the appendices and the codebase to emulate the data collection.

## 3.1   Collection and Analysis of Task Data

Our dataset `dopanim` targets a classification task with images of animal species with groups of highly similar appearances, which we call doppelgangers. Hence, accurate annotations require domain knowledge and a high level of attention. The images originate from iNaturalist [21], a platform whose observers contribute biodiversity observations. iNaturalist is particularly suitable as an image source because it features an extensive, diverse collection of high-quality images across the globe. Each observation we collected from iNaturalist is licensed as CC0, CC-BY, CC-BY-SA, CC-BY-NC, or CC BY-NC-SA and includes metadata such as location and time. The associated images span 15 classes across four animal groups, each containing doppelganger animals. Following the approach of Van Horn et al. [57], the training, validation, and test splits contain no overlap between observers to avoid learning observer-specific characteristics[1]. Research-grade images, validated by multiple iNaturalist users, ensure reliable ground truth class labels [12] and, thus, the dataset's reliability for evaluation. We demonstrate the animals' doppelganger characteristics within a group by fine-tuning a DINOv2

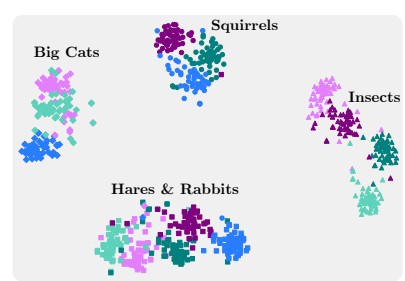

Figure 2: $t$-SNE of validation images' embeddings from a DINOv2 ViT-S/14 fine-tuned on `dopanim`.

vision transformer (ViT) [34] on training data with ground truth labels. The learned features are reduced to a two-dimensional space via $t$-distributed stochastic neighborhood embedding ($t$-SNE) [56], depicting validation animal images as markers in Fig. 2. While the four groups (marker shapes) are clearly distinguishable, animal classes within a group (marker colors) exhibit high similarities.

## 3.2   Collection and Analysis of Annotation Data

We organized the annotation data collection into several steps. Initially, student assistants were recruited as annotators. Each annotator received a basic tutorial on using LabelStudio [55] as the annotation platform, including an overview of the annotation interface and an example image for each of the 15 animal classes. Additionally, certain annotators received advanced tutorials on specific animal groups to ensure different levels of expertise. After studying the tutorials, annotators completed a pre-questionnaire with self-assessment and technical questions about wildlife. The subsequent annotation tasks involved assigning unnormalized label likelihoods to images, as shown in Fig. 3. Once the annotators had completed their annotation tasks, they filled out a post-questionnaire with self-assessment questions about the difficulties they encountered during the annotation process.

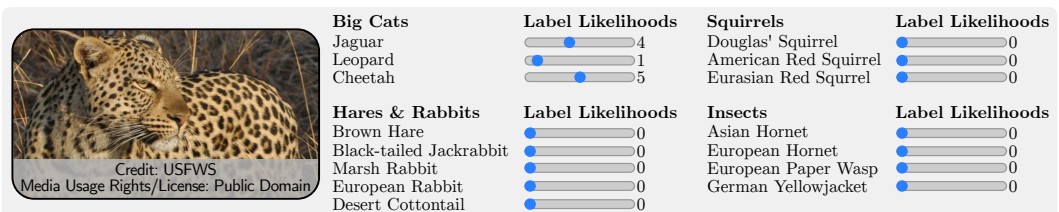

Figure 3: Annotation interface – Annotators adjust sliders for different classes to set their label likelihoods. A slider value represents the relative likelihood of an image belonging to a specific class compared to others. The label likelihoods' absolute values are unimportant; only their comparison matters. A label likelihood of zero indicates certainty that the image does not belong to that class. If there is uncertainty about the ground truth class, non-zero likelihoods can be set for multiple classes.

---

[1]This splitting resulted in updated validation and test sets for the camera-ready version, leading to slightly different empirical results without introducing substantial changes.

The collected annotation data comprises four main components: (1) tutorials, (2) pre-questionnaire, (3) post-questionnaire, and (4) individual image annotations. Components (1)-(3) allow us to extract annotator metadata, which captures task-related information about the annotators, e.g., the interest in wildlife. In addition to the assigned unnormalized likelihoods, component (4) includes the required annotation times and timestamps. All this annotation data is detailed in the appendices and can be used to evaluate various learning scenarios (cf. Section 5 for corresponding use cases). For example, we can determine the top-label predictions from the likelihoods to compute the confusion matrix in Fig. 4. We can see that animal classes are mainly confused within a group of doppelganger animals.

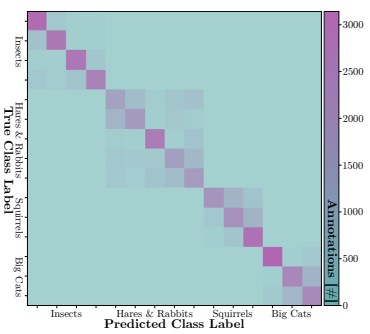

Figure 4: Confusion matrix across all human top-label predictions.

# 4 Benchmark: Multi-annotator Learning

This section presents a benchmark of common approaches in multi-annotator learning [68], also called learning from crowds [39]. We briefly outline this research area's foundations before presenting seven variants of dopanim as the basis for the experimental setup, empirical results, and future research. Our appendices further detail this benchmark, e.g., by listing computational resources.

## 4.1 Foundations

Multi-annotator learning approaches typically assume independently working annotators and require knowing which annotation originates from which annotator. This information allows them to estimate annotators' individual performances for correcting noisy class labels. Major differences in the approaches arise in their assumptions about these performances and their training procedures [17].

Table 2: Overview of one-stage multi-annotator learning approaches – The first column lists the names of the approaches, with the following columns detailing relevant attributes for each approach.

| Approach | Venue | Year | Annotator Performance Model | Training | Metadata |
|---|---|---|---|---|---|
| _Class-dependent Annotator Performances_ | | | | | |
| cl [43] | AAAI | 2018 | noise adaption layer per annotator | cross-entropy | ✗ |
| trace-reg [53] | CVPR | 2019 | confusion matrix per annotator | cross-entropy + regularization | ✗ |
| conal [7] | AAAI | 2021 | noise adaption layer per and across annotators | cross-entropy + regularization | ✓ |
| union-net [61] | TNNLS | 2022 | noise adaption layer across annotators | cross-entropy | ✗ |
| geo-reg-w [20] geo-reg-f [20] | ICLR | 2023 | confusion matrix per annotator | cross-entropy + regularization | ✗ |
| _Instance-dependent Annotator Performances_ | | | | | |
| madl [17] | TMLR | 2023 | confusion matrix per instance-annotator pair | cross-entropy + regularization | ✓ |
| crowd-ar [4] | SIGIR | 2023 | reliability scalar per instance-annotator pair | two-model cross-entropy | ✓ |
| annot-mix [18] | ECAI | 2024 | confusion matrix per instance-annotator pair | cross-entropy + mixup extension | ✓ |

_Annotation Performance Assumptions._ The simplest assumption is that an annotator's performance is constant across all classes and instances, often represented by a single accuracy score per annotator [44]. However, this assumption is unrealistic, e.g., due to varying difficulty levels across different classes and instances. Therefore, performance is often modeled with class dependency, typically by estimating a confusion matrix for each annotator [26, 53]. This matrix captures the conditional probability of an annotator assigning a certain class label, conditioned on the actual class of the instance. Some approaches also incorporate instance dependency, which considers the variability in annotator performance based on specific instances [74, 45]. For example, annotators may perform better in certain regions of the feature space. However, this most realistic assumption comes at the cost of increased training complexity.

_Training Procedures._ Training procedures are often divided into one-stage and two-stage procedures [30]. _Two-stage_ procedures aggregate multiple class labels per instance as estimates of the ground truth class labels in the first stage. These aggregated labels are then used for standard supervised learning in the second stage. The label aggregation is implemented via ground truth inference algorithms [73]. The simplest algorithm is majority voting, specifying an instance's class label as

the one with the most annotator votes. Thereby, majority voting assumes annotators have equal performance [6]. Advanced ground truth inference algorithms [9, 64, 54] drop this naive assumption by estimating annotators' performances for label aggregation. As an alternative to aggregated hard labels, soft majority voting normalizes the annotators' votes across the potential classes to obtain a probabilistic label vector. Further, training can be restricted to instances with strong agreement among annotators' class labels. All these algorithms typically require multiple class labels per instance [26], leading to high annotation costs. *One-stage* training procedures overcome this requirement, enabling learning with just one class label per instance. Typically, this involves training two models: a classification model and an annotator performance model [17]. Early one-stage procedures leverage the expectation-maximization (EM) algorithm, where the E-step estimates the ground truth labels as latent variables, and the M-step updates the models based on these estimates [39, 1]. State-of-the-art procedures use NN-based end-to-end systems, coupling both models' outputs into a single loss function for simultaneous training [7, 20, 18].

## 4.2 Dataset Variants

We create seven dataset variants (annotation subsets) of `dopanim` to test different learning scenarios. Two critical variables in multi-annotator learning are annotator performance and the number of annotations per instance [47]. We simulate varying annotator performance levels by either `randomly` selecting annotations per instance or by selecting the `worst` (false if available) annotations per instance [62]. Further, we control the number of annotations per instance, testing scenarios with very few (1 or 2), a variable number (`v`), and many (`full`) annotations per instance. Table 3 summarizes the seven variants' statistics. For future research, dataset users are free to create additional variants.

Table 3: Overview of `dopanim` variants – Column headings indicate the names of the dataset variants, whereas a row provides statistics about an annotation data characteristic of the respective variants. We denote absolute numbers by the # symbol. Means are supplemented by standard deviations.

| `dopanim` **Variant** | `worst-1` | `worst-2` | `worst-v` | `rand-1` | `rand-2` | `rand-v` | `full` |
|---|---|---|---|---|---|---|---|
| | Annotation Data | | | | | | |
| annotations per instance [#] | $1.0_{\pm 0.0}$ | $2.0_{\pm 0.0}$ | $3.0_{\pm 1.4}$ | $1.0_{\pm 0.0}$ | $2.0_{\pm 0.0}$ | $3.0_{\pm 1.4}$ | $5.0_{\pm 0.2}$ |
| annotations per annotator [#] | $524_{\pm 296}$ | $1,048_{\pm 525}$ | $1,555_{\pm 721}$ | $524_{\pm 256}$ | $1,048_{\pm 512}$ | $1,564_{\pm 746}$ | $2,602_{\pm 1,255}$ |
| overall accuracy [%] | 22.4 | 37.3 | 54.8 | 67.5 | 67.2 | 67.3 | 67.3 |
| majority voting accuracy [%] | 22.4 | 37.8 | 53.1 | 67.5 | 67.1 | 73.7 | 80.7 |
| accuracy per annotator [%] | $25.8_{\pm 12.7}$ | $39.4_{\pm 15.4}$ | $54.5_{\pm 16.1}$ | $65.2_{\pm 14.6}$ | $65.6_{\pm 14.8}$ | $65.6_{\pm 14.7}$ | $65.6_{\pm 14.7}$ |

## 4.3 Experimental Setup

*Approaches.* We consider state-of-the-art approaches from the literature. Thus, we focus on one-stage end-to-end approaches, which train NNs by estimating annotator performances to counteract noisy annotations. Table 2 overviews the corresponding approaches. For transparency, we note that the approaches `madl` [17] and `annot-mix` [18] were proposed by our research group in previous works. As an upper baseline, we evaluate the training with the ground truth class labels (`gt-base`). As lower baselines, we train with (hard) majority vote class labels (`mv-base`), with soft majority vote labels (`smv-base`), and only on instances with (hard) majority vote labels achieving at least 70% annotation agreement (`sf-base`). The latter baseline is inspired by the concept of selection frequency [40].

*Evaluation Scores.* For quantitative evaluation, we employ three scores suitable for balanced classification problems. Accuracy (`acc`) evaluated on the test data is the most common measure of generalization performance. Since many applications require more than just the actual class prediction, we also assess the quality of the predicted class probabilities. Specifically, we employ the Brier score [3] (`bs`) as a proper scoring rule and the top-label calibration error [28] (`tce`) with a more intuitive interpretation.

*Architecture.* With the advance of self-supervised learning [24], numerous pre-trained model architectures for image data are available, making training from scratch necessary only in special application domains. Thus, we use a pre-trained DINOv2 ViT-S/14 [34] as our backbone. The training of the multi-annotator

Table 4: Hyperparameters.

| **Hyperparameter** | **Value** |
|---|---|
| Architecture | |
| backbone | DINOv2 ViT-S/14 |
| classification head | MLP |
| Training | |
| optimizer | RAdam |
| learning rate scheduler | cosine annealing |
| number of epochs | 50 |
| initial learning rate | 1e-3 |
| batch size | 64 |
| weight decay | 0 |
| dropout rate | 0.5 |

learning approaches is then implemented by fine-tuning the backbone's classification head in the form of a multi-layer perceptron (MLP) with 128 hidden neurons, batch normalization [22], dropout [52], and rectified linear units (ReLU) [14] as activation function.

*Training.* We employ RAdam [31] as the optimizer across all dataset variants and multi-annotator learning approaches. Further, we schedule the learning rate over 50 training epochs via cosine annealing [32]. Concrete values for the other hyperparameters, i.e., initial learning rate, batch size, and weight decay are determined by optimizing the validation accuracy of the upper baseline `gt-base` and are reported in Table 4. Hyperparameters specific to each approach are defined according to the authors' recommendations. This way, we aim to avoid biasing results in favor of or against any particular approach, allowing for a fair empirical comparison [62]. Each training is repeated ten times with different random initializations of the NNs' parameters per dataset variant and approach. All results are reported as means and standard deviations over the ten repetitions of the respective experiment.

## 4.4 Empirical Results

Table 5 reports the individual results for all approaches, dataset variants, and evaluation scores. A more compact overview of these results (excluding the upper baseline `gt-base`) is given in the form of a ranking by Fig. 5. Lower ranks indicate better evaluation scores. As expected, the tabular results confirm that training with the ground truth class labels (`gt-base`) is superior across all dataset variants and evaluation scores. Learning from the majority vote (`mv-base`) and soft majority vote (`smv-base`) labels as lower baselines leads on average to inferior accuracy results in comparison with all other approaches, whereas the naive idea of ignoring instances with ambiguous annotations (`sf-base`) performs partially better than the worst one-stage multi-annotator learning approaches. Nevertheless, there are several one-stage multi-annotator learning approaches, e.g., `geo-reg-w`, `geo-reg-f`, and `annot-mix`, achieving noticeable accuracy and Brier score improvements compared to the lower baselines. In contrast, the top-label calibration errors of `mv-base` are partially competitive or even superior, suggesting room for improving the one-stage multi-annotator learning approaches' calibration. Approaches modeling instance-dependent annotator performances are not consistently better than approaches modeling only class-dependent annotator

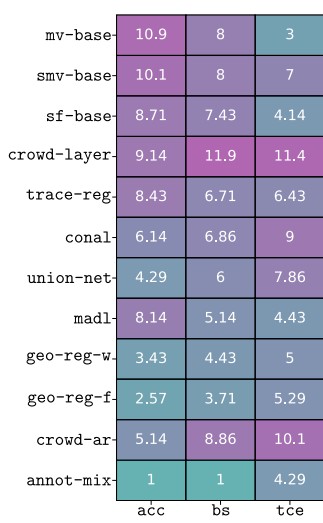

Figure 5: Mean ranks (↓) across the seven dataset variants.

performances, likely due to the more complex training of annotator performance models. Overall, the most superior approach is `annot-mix` [18] using a `mixup` [72] extension, followed by `geo-reg-f` [20] with a regularized loss function.

> *Takeaway:* The empirical results on the seven variants of `dopanim` demonstrate the potential benefit of using one-stage multi-annotator learning approaches to counteract annotation noise.

# 5 Further Use Cases

This section discusses three illustrative evaluation use cases to demonstrate our dataset's potential for exploring further research areas of machine learning.

## 5.1 Beyond Hard Class Labels

*Foundations.* Instead of forcing hard decisions on one class label, human annotators can assign likelihoods to express their (subjective) uncertainty about an annotation decision.

*Research Question.* Are the human-estimated likelihoods (probabilistic labels after normalization) reliable, and can they act as weights for the annotators' votes to improve the generalization performance of the lower baselines `mv-base`, `smv-base`, and `sf-base`?

*Experimental Setup.* We qualitatively evaluate the human-estimated likelihoods across all annotators through a reliability diagram [10] by normalizing them to obtain top-label probabilities. Such a reliability diagram depicts the annotation accuracy ($y$-axis) as a function of the top-label probability ($x$-axis) in the form of a calibration curve. The latter quantity can also be interpreted as the annotators' confidences, whose distribution can be visualized through a histogram. Following the experimental setup of Section 4, we perform a quantitative evaluation by leveraging the likelihoods as annotation weights for training `mv-base`, `smv-base`, and `sf-base` on our seven dataset variants.

*Empirical Results.* Figure 6 displays the reliability diagram. Although the annotators are confident in most of their decisions, there are also various annotations with low confidence. The general trend of the observed calibration curve shows that a higher top-label probability implies a higher accuracy. Yet, the annotators tend to overestimate their accuracies for annotations with very high confidence. Table 6 lists the accuracy gains of training the three lower baselines with likelihoods. We observe positive gains for almost all combinations of approaches and dataset variants. The likelihoods are particularly helpful for settings with many false annotations, as shown by the substantial gains for the three `worst` variants. The likelihoods have no impact on the performance of `mv-base` for `worst-1` and `rand-1` since the majority vote remains unchanged in the case of one annotation per instance.

> *Takeaway:* Human-estimated likelihoods as part of `dopanim` can be leveraged to counteract annotation noise.

## 5.2 Annotator Metadata

*Foundations.* Annotator metadata [75, 17] describes task-related properties of the annotators. Ideally, this information can be collected cost-effectively and is connected to annotator performance.

*Research Question.* Can multi-annotator learning approaches benefit from annotator metadata?

*Experimental Setup.* Before and after the annotation campaign, each annotator completed a questionnaire in which only task-related data was collected. Accordingly, the metadata does not reveal an annotator's identity. Moreover, we have access to information about absolved tutorials and annotation statistics for each annotator, e.g., mean annotation time and an annotator's consistency when pro-

Table 5: Results – Column headings list the multi-annotator learning approaches. Rows indicate the variants of `dopanim`. Results report the mean and standard deviation across ten repeated experiments. Arrows indicate whether the evaluation score is to be maximized ($\uparrow$) or minimized ($\downarrow$). **Best** and **second best** results are marked per dataset variant and evaluation score, while excluding `gt-base`.

| Approach | gt-base | mv-base | smv-base | sf-base | cl [43] | trace-reg [53] | conal [7] | union-net [61] | madl [17] | geo-reg-w [20] | geo-reg-f [20] | crowd-ar [4] | annot-mix [18] |
|---|---|---|---|---|---|---|---|---|---|---|---|---|---|
| | | baselines | | | | | | | | | | | |
| **Test Accuracy (acc [%]) $\uparrow$** | | | | | | | | | | | | | |
| worst-1 | $87.1_{\pm0.2}$ | $28.1_{\pm0.2}$ | $28.1_{\pm0.2}$ | $28.1_{\pm0.2}$ | $\mathbf{32.8}_{\pm0.2}$ | $28.4_{\pm0.4}$ | $28.9_{\pm0.3}$ | $30.7_{\pm0.6}$ | $27.8_{\pm1.4}$ | $30.6_{\pm0.5}$ | $30.5_{\pm0.4}$ | $29.0_{\pm0.6}$ | $\mathbf{34.7}_{\pm1.1}$ |
| worst-2 | $87.1_{\pm0.2}$ | $42.4_{\pm0.4}$ | $44.4_{\pm0.4}$ | $47.0_{\pm0.2}$ | $43.9_{\pm1.6}$ | $44.7_{\pm0.4}$ | $45.3_{\pm0.4}$ | $47.3_{\pm1.5}$ | $46.4_{\pm1.0}$ | $\mathbf{48.5}_{\pm0.7}$ | $47.6_{\pm0.3}$ | $44.8_{\pm0.3}$ | $\mathbf{49.7}_{\pm0.4}$ |
| worst-v | $87.1_{\pm0.2}$ | $58.3_{\pm0.5}$ | $55.6_{\pm0.5}$ | $56.2_{\pm0.5}$ | $61.3_{\pm2.3}$ | $64.1_{\pm0.4}$ | $64.9_{\pm0.8}$ | $66.5_{\pm0.8}$ | $65.9_{\pm1.5}$ | $66.8_{\pm0.3}$ | $\mathbf{69.4}_{\pm0.4}$ | $64.7_{\pm0.6}$ | $\mathbf{73.2}_{\pm0.6}$ |
| rand-1 | $87.1_{\pm0.2}$ | $71.9_{\pm0.2}$ | $71.9_{\pm0.2}$ | $71.9_{\pm0.2}$ | $73.8_{\pm3.2}$ | $72.0_{\pm0.3}$ | $75.9_{\pm0.2}$ | $76.3_{\pm1.0}$ | $71.7_{\pm0.8}$ | $76.3_{\pm0.3}$ | $\mathbf{76.9}_{\pm0.3}$ | $76.8_{\pm0.2}$ | $\mathbf{78.2}_{\pm0.4}$ |
| rand-2 | $87.1_{\pm0.2}$ | $72.5_{\pm0.5}$ | $74.9_{\pm0.2}$ | $77.0_{\pm0.2}$ | $73.1_{\pm3.0}$ | $75.0_{\pm0.2}$ | $78.5_{\pm0.3}$ | $79.3_{\pm0.4}$ | $75.5_{\pm0.7}$ | $79.0_{\pm0.5}$ | $\mathbf{79.6}_{\pm0.2}$ | $79.1_{\pm0.4}$ | $\mathbf{80.0}_{\pm0.3}$ |
| rand-v | $87.1_{\pm0.2}$ | $74.2_{\pm0.3}$ | $75.0_{\pm0.4}$ | $75.0_{\pm0.3}$ | $73.6_{\pm1.7}$ | $76.6_{\pm0.3}$ | $79.3_{\pm0.2}$ | $79.4_{\pm1.6}$ | $77.4_{\pm2.0}$ | $80.1_{\pm0.4}$ | $\mathbf{80.6}_{\pm0.3}$ | $79.9_{\pm0.3}$ | $\mathbf{81.3}_{\pm0.3}$ |
| full | $87.1_{\pm0.2}$ | $77.8_{\pm0.3}$ | $78.4_{\pm0.5}$ | $78.9_{\pm0.3}$ | $73.9_{\pm3.8}$ | $78.6_{\pm0.5}$ | $80.3_{\pm0.2}$ | $80.1_{\pm1.6}$ | $79.6_{\pm1.8}$ | $80.9_{\pm0.5}$ | $\mathbf{81.4}_{\pm0.2}$ | $80.9_{\pm0.2}$ | $\mathbf{82.2}_{\pm0.2}$ |
| **Test Brier Score (bs) $\downarrow$** | | | | | | | | | | | | | |
| worst-1 | $0.19_{\pm.00}$ | $\mathbf{0.91}_{\pm.00}$ | $0.91_{\pm.00}$ | $0.91_{\pm.00}$ | $1.26_{\pm.01}$ | $0.92_{\pm.01}$ | $1.20_{\pm.01}$ | $1.23_{\pm.01}$ | $0.96_{\pm.02}$ | $1.17_{\pm.01}$ | $1.18_{\pm.01}$ | $1.26_{\pm.02}$ | $\mathbf{0.87}_{\pm.02}$ |
| worst-2 | $0.19_{\pm.00}$ | $0.73_{\pm.00}$ | $\mathbf{0.68}_{\pm.00}$ | $0.74_{\pm.00}$ | $1.01_{\pm.02}$ | $0.68_{\pm.00}$ | $0.90_{\pm.01}$ | $0.87_{\pm.03}$ | $0.68_{\pm.02}$ | $0.82_{\pm.01}$ | $0.83_{\pm.01}$ | $0.96_{\pm.01}$ | $\mathbf{0.63}_{\pm.00}$ |
| worst-v | $0.19_{\pm.00}$ | $0.55_{\pm.00}$ | $0.55_{\pm.00}$ | $0.59_{\pm.00}$ | $0.68_{\pm.05}$ | $0.47_{\pm.00}$ | $0.54_{\pm.00}$ | $0.51_{\pm.02}$ | $0.45_{\pm.02}$ | $0.49_{\pm.01}$ | $\mathbf{0.44}_{\pm.00}$ | $0.59_{\pm.01}$ | $\mathbf{0.37}_{\pm.01}$ |
| rand-1 | $0.19_{\pm.00}$ | $0.39_{\pm.00}$ | $0.39_{\pm.00}$ | $0.39_{\pm.00}$ | $0.43_{\pm.07}$ | $0.39_{\pm.00}$ | $0.37_{\pm.00}$ | $0.36_{\pm.02}$ | $0.39_{\pm.01}$ | $0.35_{\pm.00}$ | $\mathbf{0.35}_{\pm.00}$ | $0.37_{\pm.01}$ | $\mathbf{0.31}_{\pm.00}$ |
| rand-2 | $0.19_{\pm.00}$ | $0.39_{\pm.00}$ | $0.37_{\pm.00}$ | $0.33_{\pm.00}$ | $0.46_{\pm.06}$ | $0.36_{\pm.00}$ | $0.33_{\pm.00}$ | $0.31_{\pm.00}$ | $0.34_{\pm.01}$ | $0.31_{\pm.00}$ | $\mathbf{0.30}_{\pm.00}$ | $0.34_{\pm.01}$ | $\mathbf{0.29}_{\pm.00}$ |
| rand-v | $0.19_{\pm.00}$ | $0.36_{\pm.00}$ | $0.37_{\pm.00}$ | $0.35_{\pm.00}$ | $0.45_{\pm.03}$ | $0.35_{\pm.00}$ | $0.32_{\pm.00}$ | $0.31_{\pm.00}$ | $0.31_{\pm.02}$ | $0.28_{\pm.00}$ | $\mathbf{0.28}_{\pm.00}$ | $0.33_{\pm.01}$ | $\mathbf{0.27}_{\pm.00}$ |
| full | $0.19_{\pm.00}$ | $0.32_{\pm.00}$ | $0.36_{\pm.00}$ | $0.31_{\pm.00}$ | $0.44_{\pm.08}$ | $0.34_{\pm.00}$ | $0.31_{\pm.00}$ | $0.30_{\pm.04}$ | $0.29_{\pm.02}$ | $0.27_{\pm.01}$ | $\mathbf{0.27}_{\pm.00}$ | $0.31_{\pm.00}$ | $\mathbf{0.26}_{\pm.00}$ |
| **Test Top-label Calibration Error (tce) $\downarrow$** | | | | | | | | | | | | | |
| worst-1 | $0.06_{\pm.00}$ | $\mathbf{0.33}_{\pm.00}$ | $\mathbf{0.33}_{\pm.00}$ | $0.33_{\pm.00}$ | $0.61_{\pm.01}$ | $0.33_{\pm.00}$ | $0.54_{\pm.00}$ | $0.59_{\pm.02}$ | $0.37_{\pm.01}$ | $0.53_{\pm.01}$ | $0.53_{\pm.01}$ | $0.59_{\pm.01}$ | $0.34_{\pm.01}$ |
| worst-2 | $0.06_{\pm.00}$ | $0.22_{\pm.00}$ | $\mathbf{0.22}_{\pm.01}$ | $0.27_{\pm.00}$ | $0.46_{\pm.02}$ | $\mathbf{0.22}_{\pm.00}$ | $0.40_{\pm.01}$ | $0.39_{\pm.02}$ | $0.24_{\pm.01}$ | $0.35_{\pm.01}$ | $0.36_{\pm.00}$ | $0.45_{\pm.01}$ | $0.24_{\pm.00}$ |
| worst-v | $0.06_{\pm.00}$ | $0.13_{\pm.00}$ | $0.15_{\pm.01}$ | $0.15_{\pm.00}$ | $0.25_{\pm.03}$ | $0.14_{\pm.00}$ | $0.19_{\pm.01}$ | $0.16_{\pm.01}$ | $\mathbf{0.12}_{\pm.01}$ | $0.14_{\pm.01}$ | $0.14_{\pm.00}$ | $0.23_{\pm.01}$ | $\mathbf{0.11}_{\pm.00}$ |
| rand-1 | $0.06_{\pm.00}$ | $0.10_{\pm.00}$ | $0.10_{\pm.00}$ | $0.10_{\pm.00}$ | $0.15_{\pm.03}$ | $\mathbf{0.09}_{\pm.00}$ | $0.13_{\pm.00}$ | $0.12_{\pm.01}$ | $\mathbf{0.08}_{\pm.01}$ | $0.11_{\pm.00}$ | $0.11_{\pm.00}$ | $0.14_{\pm.00}$ | $0.10_{\pm.00}$ |
| rand-2 | $0.06_{\pm.00}$ | $\mathbf{0.10}_{\pm.00}$ | $0.14_{\pm.00}$ | $\mathbf{0.10}_{\pm.00}$ | $0.16_{\pm.02}$ | $0.13_{\pm.00}$ | $0.12_{\pm.00}$ | $0.10_{\pm.01}$ | $0.10_{\pm.01}$ | $0.10_{\pm.00}$ | $0.10_{\pm.00}$ | $0.14_{\pm.00}$ | $0.10_{\pm.00}$ |
| rand-v | $0.06_{\pm.00}$ | $0.09_{\pm.00}$ | $0.15_{\pm.00}$ | $\mathbf{0.09}_{\pm.00}$ | $0.15_{\pm.01}$ | $0.15_{\pm.01}$ | $0.11_{\pm.00}$ | $0.10_{\pm.01}$ | $0.10_{\pm.01}$ | $\mathbf{0.09}_{\pm.00}$ | $0.09_{\pm.00}$ | $0.13_{\pm.00}$ | $0.10_{\pm.00}$ |
| full | $0.06_{\pm.00}$ | $0.09_{\pm.00}$ | $0.20_{\pm.00}$ | $0.10_{\pm.00}$ | $0.14_{\pm.03}$ | $0.16_{\pm.00}$ | $0.11_{\pm.00}$ | $0.10_{\pm.01}$ | $0.10_{\pm.01}$ | $\mathbf{0.09}_{\pm.01}$ | $\mathbf{0.09}_{\pm.00}$ | $0.12_{\pm.00}$ | $0.10_{\pm.00}$ |

cessing the same images multiple times. All this information is summarized as a vector of metadata features per annotator. We analyze the potential benefit of these features through correlation analysis and by integrating the metadata into the training of multi-annotator learning approaches. Following the experimental setup of Section 4, we employ `madl` and `annot-mix` that process the metadata as vectorial inputs. The appendices provide more details about the individual metadata features.

*Empirical Results.* Figure 7 presents Spearman's rank correlation coefficients [51] between observed annotator accuracies and selected annotator metadata features. Notably, there is a strong correlation between annotator accuracy and both the accuracy of answers to technical wildlife questions and the consistency of annotations across identical images. Table 7 lists the accuracy gains of training `madl` and `annot-mix` with additional annotator metadata. There are clear positive gains across all dataset variants when training `madl` with annotator metadata. In contrast, `annot-mix` slightly benefits from the annotator metadata for only four of the seven variants and even results in a slight accuracy decrease once. The higher gains for `madl` can likely be attributed to improved similarity computations between embeddings of annotators as a core part of its training process.

> *Takeaway:* Annotator metadata as part of `dopanim` can be leveraged to improve the training of multi-annotator learning approaches.

## 5.3 Annotation Times in Active Learning

*Foundations.* Active learning [42] strategies aim to reduce the annotation cost while maximizing the generalization performance by selecting only the most useful instances for annotation. The most common strategy, uncertainty sampling [48], selects instances with the highest model uncertainty.

*Research Question.* Do instances selected by uncertainty sampling require more time for annotation?

*Experimental Setup.* We compare uncertainty sampling to random sampling as a baseline strategy. For simplicity, we use the ground truth labels to fine-tune a linear layer as the classification head of the pre-trained DINOv2 ViT-S/14 model [34]. At the start of active learning, we randomly annotate 100 images. In each subsequent iteration, the respective strategy selects 100 new images for annotation. We repeat this experiment 50 times and report means and standard deviations.

*Empirical Results.* The histogram in Fig. 8 indicates high variance in annotation times, which is also reflected in the active learning results shown in Fig. 9. Uncertainty sampling achieves higher accuracies ($y$-axis) with the same number of annotations ($x$-axis) than random sampling. However, the mean annotation time per instance selected by uncertainty sampling is about 0.5s longer.

> *Takeaway:* With `dopanim`, we demonstrate that uncertainty sampling selects images with higher annotation times.

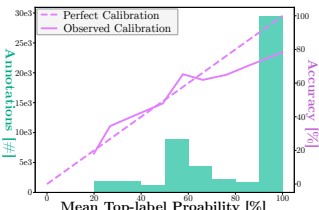

Figure 6: Calibration plot of human top-label probabilities.

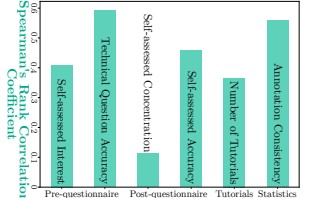

Figure 7: Correlation of annotator metadata and accuracies.

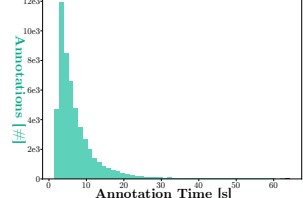

Figure 8: Histogram of all recorded annotation times.

Table 6: Probabilistic labels.

| Approach | `mv-base` | `smv-base` | `sf-base` |
|---|---|---|---|
| Test Accuracy Gains (+`acc` [%]) | | | |
| `worst-1` | $0.0_{\pm 0.0}$ | $+10.6_{\pm 0.4}$ | $+9.0_{\pm 0.3}$ |
| `worst-2` | $+7.1_{\pm 0.6}$ | $+9.5_{\pm 0.5}$ | $+7.4_{\pm 0.3}$ |
| `worst-v` | $+3.0_{\pm 0.5}$ | $+8.2_{\pm 0.6}$ | $+6.8_{\pm 0.5}$ |
| `rand-1` | $0.0_{\pm 0.0}$ | $+2.0_{\pm 0.4}$ | $+1.8_{\pm 0.4}$ |
| `rand-2` | $+1.9_{\pm 0.5}$ | $+1.9_{\pm 0.5}$ | $+0.7_{\pm 0.3}$ |
| `rand-v` | $+1.3_{\pm 0.4}$ | $+2.1_{\pm 0.5}$ | $+1.8_{\pm 0.5}$ |
| `full` | $+0.6_{\pm 0.5}$ | $+1.0_{\pm 0.4}$ | $-0.2_{\pm 0.3}$ |

Table 7: Annotator metadata.

| Approach | `madl` | `annot-mix` |
|---|---|---|
| Test Accuracy Gains (+`acc` [%]) | | |
| `worst-1` | $+3.4_{\pm 2.5}$ | $+0.7_{\pm 2.0}$ |
| `worst-2` | $+3.3_{\pm 1.3}$ | $-0.6_{\pm 0.7}$ |
| `worst-v` | $+1.5_{\pm 2.4}$ | $+0.2_{\pm 0.6}$ |
| `rand-1` | $+3.1_{\pm 0.8}$ | $+0.4_{\pm 0.4}$ |
| `rand-2` | $+1.3_{\pm 1.0}$ | $+0.1_{\pm 0.5}$ |
| `rand-v` | $+1.6_{\pm 2.2}$ | $0.0_{\pm 0.5}$ |
| `full` | $+0.2_{\pm 1.0}$ | $0.0_{\pm 0.3}$ |

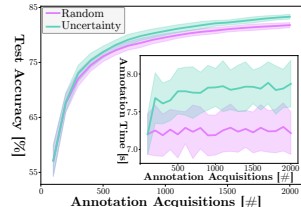

Figure 9: Active learning curves of accuracy and time.

# 6 Conclusion and Limitations

*Conclusion.* We introduced the `dopanim` dataset containing images with ground truth labels of animal species with groups of similar appearance (`doppelganger animals`). In a comprehensive annotation campaign, error-prone humans annotated this dataset using likelihoods to express their subjective uncertainty. The received annotations are supplemented with annotator metadata [75] containing task-related information collected via questionnaires, tutorials, and annotation statistics. A benchmark study evaluated the benefit of multi-annotator learning approaches [68] for seven variants of `dopanim`. Moreover, three use cases demonstrated `dopanim`'s potential for further research. Data and codebase, including multi-annotator learning approaches, backbones, experiments, and evaluation protocols, are publicly accessible, facilitating methodological research in various machine learning fields.

*Limitations.* Due to the high costs of annotations, `dopanim` is a small-scale dataset. To test scalability, more images and animal classes must be included and annotated. For this purpose, our codebase for collecting `dopanim` using iNaturalist [21] and LabelStudio [55] can be easily adapted. Additionally, our benchmark of multi-annotator learning approaches is limited to the seven variants of `dopanim`, DINOv2 ViT-S/14 [34] as the backbone architecture, and one hyperparameter configuration (specified via the validation accuracy of the upper baseline `gt-base`) per approach. However, our codebase includes other backbones and related datasets with noisy annotations from multiple humans, allowing for future benchmark extensions. Particularly, such extensions may include the evaluation of techniques for hyperparameter optimization, e.g.,

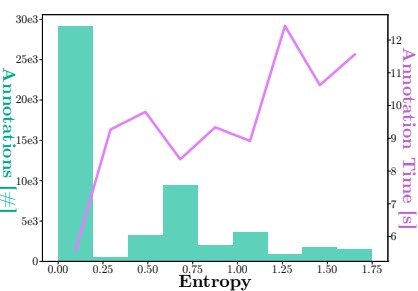

Figure 10: Annotation time as a function of entropy of normalized human-estimated likelihoods.

number of training epochs via early stopping [70], in the presence of noisy labels and approaches for learning from noisy labels [50], which do not rely on the information which label originates from which annotator. The three presented use cases also require further investigations to derive conclusive results. For example, it is still unclear whether probabilistic labels are also beneficial from a cost-sensitive perspective because Fig. 10 indicates increasing annotation times with an increasing entropy of the (normalized) human estimated likelihoods. Yet, such a trend does not directly quantify the actual amount of higher costs for querying likelihoods because also requesting hard class labels can take more time for uncertain (difficult) images. Finally, `dopanim` is a dataset targeting classification tasks where an objective ground truth exists. However, certain tasks involve subjective class labels, e.g., assessing emotions [23] or laughter [58], reflecting variations in annotators' interpretations.

## Acknowledgments and Disclosure of Funding

This work was funded by the ALDeep and CIL projects at the University of Kassel. Moreover, we thank Franz Götz-Hahn and the anonymous reviewers for their insightful comments. Finally, we thank the iNaturalist [21] community for their many observations that help explore our nature's biodiversity and our annotators for their efforts in making the annotation campaign via LabelStudio [55] possible.

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

# A Overview

The appendices provide additional information beyond the primary resources of our dataset `dopanim`, which are:

- the **main paper** presenting our core findings,
- the **dataset** on Zenodo[2] offering direct access to the data,
- and the **codebase** on GitHub[3] ensuring reproducibility and ficilating experimentation.

We structure the appendices as follows: Appendix B presents the datasheet of the `dopanim` dataset based on a standard questionnaire. Assets used within this work are listed in Appendix C. We discuss our dataset in the societal context of collecting data with the help of humans in Appendix D. Further related datasets with noisy annotations, which did not fit our main paper's focus, are detailed by Appendix E. Finally, we provide more information about the empirical evaluation in Appendix F.

Throughout these appendices, we explicitly indicate passages that address the questions from [1 (a)] to [5 (c)] in the checklist of the main paper.

# B Datasheet

In this appendix, we present the datasheet of our dataset `dopanim` by completing the questionnaire developed by Gebru et al. [13]. The purpose of such a datasheet is to provide detailed documentation that describes the various aspects of a dataset, including its motivation, composition, collection process, preprocessing, uses, distribution, and maintenance.

## B.1 Motivation

**For what purpose was the dataset created?** *Was there a specific task in mind? Was there a specific gap that needed to be filled?*

The `dopanim` dataset was created to advance methodological research in machine learning with noisy annotations from multiple annotators. Next to multi-annotator learning [68], also referred to as learning from crowds [39], concrete use cases range from learning with noisy class labels [50] and learning beyond hard class labels [69] to active learning with error-prone annotators [16].

**Who created the dataset (e.g., which team, research group) and on behalf of which entity (e.g., company, institution, organization)?** *If there is an associated grant, please provide the name of the grantor and the grant name and number.*

The dataset was created by the researchers Marek Herde, Denis Huseljic, and Lukas Rauch of the *Intelligent Embedded Systems* (IES) group at the University of Kassel under the supervision of Bernhard Sick.

**Who funded the creation of the dataset?**

This work was funded by the ALDeep and CIL projects, with the University of Kassel as the exclusive project sponsor.

**Any other comments?**

We refer to the main paper for a more detailed description of our dataset's motivation and objectives.

## B.2 Composition

**What do the instances that comprise the dataset represent (e.g., documents, photos, people, countries)?** *Are there multiple types of instances (e.g., movies, users, and ratings; people and interactions between them; nodes and edges)? Please provide a description.*

The dataset comprises instances in the form of animal images, which are supplemented by task data (cf. Table 8), annotation data (cf. Table 9), and annotator metadata (cf. Table 10).

---

[2]`https://doi.org/10.5281/zenodo.11479589`
[3]`https://github.com/ies-research/multi-annotator-machine-learning/tree/dopanim`

Table 8: `task_data.json` – Column headings indicate the name of the data field, the data type, and the description of the values' meaning regarding the task data collected from iNaturalist [21]. Images and their ground truth labels can be extracted from `task_data.json` through our codebase.

| Data Field | Data Type | Description |
|---|---|---|
| Index | | |
| observation_id | int | unique identifier of the observation on iNaturalist |
| Wildlife | | |
| taxon_name | string | name of the taxon on iNaturalist (ground truth class label) |
| taxon_id | int | unique identifier of the taxon name on iNaturalist |
| exact_taxon_name | string | more detailed name of the taxon on iNaturalist |
| exact_taxon_id | int | unique identifier of the more detailed taxon name on iNaturalist |
| place_guess | string | name of the user's guessed place where the observation was recorded |
| location_x | float | longitude coordinate of the guessed place |
| location_y | float | latitude coordinate of the guessed place |
| captive | boolean | flag whether the observation is in captivity |
| user_id | int | unique identifier of the user who recorded the observation |
| species_guess | string | user's guess of the taxon |
| split | string | assigns the observation to the training, validation, or test split |
| Resources and Licenses | | |
| uri | string | link to the observation on iNaturalist |
| license_code | string | license code the user applied to their observation |
| photo_url | string | link to the photo on iNaturalist |
| photo_id | int | unique identifier of the photo on iNaturalist |
| photo_attribution | string | text to properly attribute the photo according to its license code |
| photo_license_code | string | license code of the photo |
| Quality | | |
| quality_grade | string | iNaturalist quality degree of the image's species |
| identifications_count | int | number of users who annotated the image on iNaturalist |
| identifications_most_agree | boolean | flag indicating whether most users agreed on the species |
| identifications_most_disagree | boolean | flag indicating whether most users disagreed on the species |
| identifications_some_agree | boolean | flag indicating whether at least two users agreed on the species |
| num_identifications_agreements | int | number of agreements between users who annotated the image on iNaturalist |
| num_identifications_disagreements | int | number of disagreements between users who annotated the image on iNaturalist |

**How many instances are there in total (of each type, if appropriate)?**

The dataset includes around 15,734 animal images (records of task data). Out of these, 10,484 images have been annotated by 20 human annotators (records of annotator metadata), resulting in a total of 52,795 annotations (records of annotation data).

**Does the dataset contain all possible instances or is it a sample (not necessarily random) of instances from a larger set? If the dataset is a sample, then what is the larger set? Is the sample representative of the larger set (e.g., geographic coverage)?** *If so, please describe how this representativeness was validated/verified. If it is not representative of the larger set, please describe why not (e.g., to cover a more diverse range of instances, because instances were withheld or unavailable)*

The dataset comprises a sample of images taken from iNaturalist [21], focusing on 15 animal classes belonging to four groups of doppelganger animals that are challenging to differentiate. As a result, these images are suitable for studying challenging classification tasks (requiring expertise for correct annotation) in the field of learning from noisy annotations [50].

**What data does each instance consist of?** *"Raw" data (e.g., unprocessed text or images) or features? In either case, please provide a description*

Each image is a `.jpeg` file named according to the unique observation identifier on iNaturalist. The task data contains additional information for each image, e.g., the location (cf. Table 8). A record in the annotation data consists next to the likelihoods of further information about the annotation process, e.g., annotation time (cf. Table 9). Task-related information about the annotators, e.g., self-estimated accuracy, acquired via questionnaires form a record in our annotator metadata (cf. Table 10).

**Is there a label or target associated with each instance?** *If so, please provide a description.*

Each image has a single ground truth class label obtained from iNaturalist (cf. `taxon_name` in Table 8). Furthermore, about five annotations in the form of unnormalized class-label likelihoods estimated by human annotators are assigned to each image (cf. `likelihoods` in Table 9).

Table 9: `annotation_data.json` – Column headings indicate the name of the data field, the data type, and the description of the values' meaning regarding the annotation data collected during the annotation campaign. Likelihoods or top-label predictions can be extracted from `annotation_data.json` through our codebase. As an additional remark, we note that the yellow-legged hornet is also known as the Asian hornet.

| Data Field | Data Type | Description |
|---|---|---|
| Index | | |
| annotation_id | int | unique identifier of the annotation |
| General | | |
| observation_id | int | unique identifier of the observation on iNaturalist |
| annotator_id | string | unique pseudonym as the identifier of the annotator |
| annotation_time | float | time it took for the annotator to provide an annotation |
| annotation_timestamp | string | date and time when the annotation was provided |
| likelihoods | dictionary | likelihoods provided by the annotator |
| Likelihoods (cf. Fig. 12) | | |
| European Hornet | float | |
| European Paper Wasp | float | |
| German Yellowjacket | float | |
| Yellow-legged Hornet | float | |
| Black-tailed Jackrabbit | float | |
| Brown Hare | float | |
| Desert Cottontail | float | |
| European Rabbit | float | likelihood value assigned to the respective animal class |
| Marsh Rabbit | float | |
| American Red Squirrel | float | |
| Douglas' Squirrel | float | |
| Eurasian Red Squirrel | float | |
| Cheetah | float | |
| Jaguar | float | |
| Leopard | float | |

**Is any information missing from individual instances?** *If so, please provide a description, explaining why this information is missing (e.g., because it was unavailable). This does not include intentionally removed information, but might include, e.g., redacted text.*

Aside from the fact that not every annotator processed every image due to the high annotation workload and limited budget, the dataset is complete.

**Are relationships between individual instances made explicit (e.g., users' movie ratings, social network links)?** *If so, please describe how these relationships are made explicit.*

The relationships between images, task data, annotation data, and annotator metadata are made explicit by using corresponding identifiers. More concretely, images are named according to the `observation_id` in the task data (cf. Table 8). This unique identifier is also part of the annotation data (cf. Table 9) to map annotations to images. Likewise, the `annotator_id` in the annotator metadata (cf. Table 10) creates a mapping to the annotation data.

**Are there recommended data splits (e.g., training, development/validation, testing)?** *If so, please provide a description of these splits, explaining the rationale behind them.*

[3 (b)] To ensure reproducibility, we provide fixed training, validation, and test splits (cf. data field `split` in Table 8), each with a balanced class distribution.

**Are there any errors, sources of noise, or redundancies in the dataset?** *If so, please provide a description.*

The annotations in our dataset were collected to capture human annotation noise. Additionally, each annotator processed several images twice to assess the consistency of their annotations.

**Is the dataset self-contained, or does it link to or otherwise rely on external resources (e.g., websites, tweets, other datasets)?** *If it links to or relies on external resources, a) are there guarantees that they will exist, and remain constant, over time; b) are there official archival versions of the complete dataset (i.e., including the external resources as they existed at the time the dataset was created); c) are there any restrictions (e.g., licenses, fees) associated with any of the external resources that might apply to a dataset consumer? Please provide descriptions of all external resources and any restrictions associated with them, as well as links or other access points, as appropriate.*

Table 10: `annotator_metadata.json` – Column headings indicate the name of the data field, the data type, and the description of the values' meaning regarding the annotator metadata. The listed fields contain only the annotator metadata collected from the pre-, post-questionnaire, and tutorials. Additional statistical annotator metadata, e.g., mean annotation time and annotation consistency, can be extracted from `annotation_data.json` through our codebase.

| Data Field | Data Type | Description |
|---|---|---|
| **Index** | | |
| annotator_id | string | unique pseudonym as the identifier of the annotator |
| **Pre-questionnaire (cf. Fig. 11)** | | |
| pre_interest_choice | string | self-assessed interest (Likert scale) in wildlife and animals |
| pre_knowledge_choice | string | self-assessed knowledge (Likert scale) about wildlife and animals |
| pre_oldest_animal_choice | string | answer to the technical question about vertebrates' lifespan |
| pre_mammal_migration_choice | string | answer to the technical question about terrestrial mammals' migration |
| pre_hares_choice | string | answer to the question about hares and rabbits |
| pre_insects_choice | string | answer to the technical question about insects |
| pre_big_cats_choice | string | answer to the technical question about big cats |
| pre_squirrels_choice | string | answer to the technical question about squirrels |
| pre_time | float | time required to complete the pre-questionnaire |
| **Post-questionnaire (cf. Fig. 13)** | | |
| post_estimated_accuracy | float | self-estimated accuracy of the top-label prediction |
| post_motivation_choice | string | self-estimated level (Likert scale) of the average motivation |
| post_tutorial_choice | string | self-estimated frequency (Likert scale) for looking at the tutorial(s) |
| post_likelihood_choice | string | self-estimated average quality level (Likert scale) for the likelihoods |
| post_concentration_choice | string | self-estimated level (Likert scale) of the average concentration |
| post_american_red_squirrel_choice | string | |
| post_asian_hornet_choice | string | |
| post_black_tailed_jackrabbit_choice | string | |
| post_brown_hare_choice | string | |
| post_cheetah_choice | string | |
| post_desert_cottontail_choice | string | |
| post_douglas_squirrel_choice | string | self-estimated average difficulty (Likert scale) for |
| post_eurasian_red_squirrel_choice | string | correctly identifying animals of the respective class |
| post_european_hornet_choice | string | |
| post_european_paper_wasp_choice | string | |
| post_european_rabbit_choice | string | |
| post_german_yellowjacket_choice | string | |
| post_jaguar_choice | string | |
| post_leopard_choice | string | |
| post_marsh_rabbit_choice | string | |
| post_time | float | time required to complete the post-questionnaire |
| **Tutorials** | | |
| basic_tutorial | float | basic tutorial quality with the possible values .0, .5, and 1. |
| big_cats_tutorial | float | flag whether an annotator got a big cats tutorial (0.) or not (1.) |
| hares_rabbits_tutorial | float | flag whether an annotator got a hares and rabbits tutorial (0.) or not (1.) |
| insects_tutorial | float | flag whether an annotator got an insects tutorial (0.) or not (1.) |
| squirrels_tutorial | float | flag whether an annotator got a squirrels tutorial (0.) or not (1.) |

The dataset is self-contained and only provides links to the observations and images for reference.

**Does the dataset contain data that might be considered confidential (e.g., data that is protected by legal privilege or by doctor-patient confidentiality, data that includes the content of individuals' non-public communications)?** *If so, please provide a description.*

[4 (e)] No (cf. Appendix D for further explanation).

**Does the dataset contain data that, if viewed directly, might be offensive, insulting, threatening, or might otherwise cause anxiety?**

iNaturalist captures the diversity of all wildlife, which includes images of deceased animals. This is also true for a few instances in our dataset. Annotators were given a trigger warning and could withdraw from the annotation campaign at any time.

**Does the dataset identify any subpopulations (e.g., by age, gender)** *If so, please describe how these subpopulations are identified and provide a description of their respective distributions within the dataset.*

[4 (e)] No (cf. Appendix D for further explanation).

**Is it possible to identify individuals (i.e., one or more natural persons), either directly or indirectly (i.e., in combination with other data) from the dataset?** *If so, please describe how*

[4 (e)] No (cf. Appendix D for further explanation).

**Does the dataset contain data that might be considered sensitive in any way (e.g., data that reveals race or ethnic origins, sexual orientations, religious beliefs, political opinions or union memberships, or locations; financial or health data; biometric or genetic data; forms of government identification, such as social security numbers; criminal history)?** *If so, please provide a description.*

[4 (e)] No (cf. Appendix D for further explanation).

**Any other comments?**

None.

## B.3 Collection Process

**How was the data associated with each instance acquired?** *Was the data directly observable (e.g., raw text, movie ratings), reported by subjects (e.g., survey responses), or indirectly inferred/derived from other data (e.g., part-of-speech tags, model-based guesses for age or language)? If the data was reported by subjects or indirectly inferred/derived from other data, was the data validated/verified? If so, please describe how.*

Our data was collected in two phases. In the first phase, we downloaded observations with corresponding images from iNaturalist. This data was then split into training, validation, and test sets. The training data was further divided into ten batches of annotation tasks, which were manually assigned to different annotators. Before starting the annotation process, each annotator completed one or more tutorials with annotation instructions and information on distinguishing between the 15 animal classes. Additionally, each annotator filled out a pre-questionnaire (cf. Fig. 11). During the annotation tasks, annotators could assign likelihoods and mark invalid images (cf. Fig. 12). After processing all assigned annotations, a post-questionnaire (cf. Fig. 13) was filled out by each annotator.

Figure 11: Pre-questionnaire [5 (a)] – Before starting with the actual annotation tasks, each annotator filled out a pre-questionnaire with questions regarding their interest in and knowledge of wildlife and animals. There were self-assessment questions on the one hand and technical questions with correct and incorrect answers on the other hand. For each question, only one of the answer options could be clicked. The pre-questionnaire was completed by clicking the submit button.

**What mechanisms or procedures were used to collect the data** *(e.g., hardware apparatuses or sensors, manual human curation, software programs, software APIs)? How were these mechanisms or procedures validated?*

We used pyinaturalist [33] as a Python interface for the iNaturalist API to download the observations and images. We organized the annotation campaign using Label Studio [55], with a community

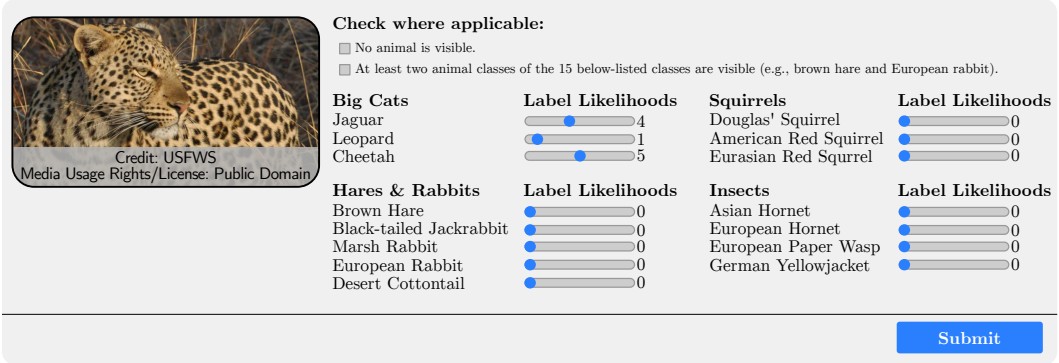

Figure 12: Annotation interface [5 (a)] – Annotators adjusted sliders for different classes to set their label likelihoods. These slider values represent the relative likelihood of an image belonging to a specific class compared to others. The label likelihoods' absolute values are unimportant; only their comparison matters. A label likelihood of zero indicates certainty that the image does not belong to that class. If there was uncertainty about the ground truth class, non-zero likelihoods could be set for multiple classes. Next to the likelihood sliders, annotators could mark images where no animal is visible or images that would qualify for a multi-label classification task. These markings were used to filter invalid images. Each annotation was completed by clicking the submit button.

edition instance set up for each annotator on an internal workstation of our university. The correctness of all implementations was ensured through preliminary tests. Code to emulate our data collection is available at our GitHub repository.

**If the dataset is a sample from a larger set, what was the sampling strategy** *(e.g., deterministic, probabilistic with specific sampling probabilities)?*

The dataset is a small random sample of 15 animal classes downloaded from iNaturalist.

**Who was involved in the data collection process** *(e.g., students, crowdworkers, contractors) and how were they compensated (e.g., how much were crowdworkers paid)?*

We managed the collection of this dataset and employed 20 student assistants making valuable contributions as annotators. Appendix D provides further details regarding their compensation.

**Over what timeframe was the data collected?** *Does this timeframe match the creation timeframe of the data associated with the instances (e.g., recent crawl of new news articles)? If not, please describe the timeframe in which the data associated with the instances was created.*

Downloading the observations and images took approximately two to three days, while the annotation campaign spanned over four months.

**Did you collect the data from the individuals in question directly, or obtain it via third parties or other sources (e.g., websites)?**

We collected all data from the annotators using Label Studio.

**Were the individuals in question notified about the data collection?** *If so, please describe (or show with screenshots or other information) how notice was provided, and provide a link or other access point to, or otherwise reproduce, the exact language of the notification itself.*

Due to the applicable licenses (cf. Table 11), no notification was sent to the iNaturalist users as data creators of observations and images. However, the annotators were explicitly notified about the usage of their annotation data (cf. Appendix D).

**Did the individuals in question consent to the collection and use of their data?** *If so, please describe (or show with screenshots or other information) how consent was requested and provided, and provide a link or other access point to, or otherwise reproduce, the exact language to which the individuals consented.*

[4 (d)] No consent was required to collect data from iNaturalist due to the applicable licenses (cf. Table 11). However, the annotators were explicitly asked for their consent (cf. Appendix D).

Figure 13: Post-questionnaire [5 (a)] – After completing the actual annotation tasks, each annotator filled out a post-questionnaire with questions regarding their experiences throughout the annotation process. There were only self-assessment questions, and only one of the answer options could be clicked. The post-questionnaire was completed by clicking the submit button.

**If consent was obtained, were the consenting individuals provided with a mechanism to revoke their consent in the future or for certain uses?** *If so, please provide a description, as well as a link or other access point to the mechanism (if appropriate).*

Yes (cf. Appendix D).

**Has an analysis of the potential impact of the dataset and its use on data subjects (e.g., a data protection impact analysis) been conducted?** *If so, please provide a description of this analysis, including the outcomes, as well as a link or other access point to any supporting documentation.*

No data protection impact analysis was conducted; however, all annotators were informed beforehand about the potential tasks and outcomes. Additionally, all steps complied with the General Data Protection Regulation (GDPR) [41]. More information is provided in Appendix D.

**Any other comments?**

None.

### B.4 Preprocessing/cleaning/labeling

**Was any preprocessing/cleaning/labeling of the data done (e.g., discretization or bucketing, tokenization, part-of-speech tagging, SIFT feature extraction, removal of instances, processing of missing values)?** *If so, please provide a description. If not, you may skip the remaining questions in this section.*

After downloading the observations and images, we filtered with the help of the annotators (cf. Fig. 12) invalid images showing no animals or depicting at least two animals of the 15 classes (multi-label).

**Was the "raw" data saved in addition to the preprocessed/cleaned/labeled data (e.g., to support unanticipated future uses)?** *If so, please provide a link or other access point to the "raw" data.*

The invalid images were manually reviewed by us, the dataset managers, to ensure they were correctly tagged as invalid. As a result, there is no value in retaining these images. All remaining data is available as part of our dataset.

**Is the software that was used to preprocess/clean/label the data available?** *If so, please provide a link or other access point.*

All tools employed to collect the dataset are freely available and are described in Appendix C.

**Any other comments?**

None

### B.5 Uses

**Has the dataset been used for any tasks already?** *If so, please provide a description.*

Except for the presented usages in the main paper, the dataset has not been employed for further tasks until now.

**Is there a repository that links to any or all papers or systems that use the dataset?** *If so, please provide a link or other access point.*

Since the dataset is new, it has not yet been used in other papers. However, we will maintain a list in our GitHub repository that provides a corresponding overview with brief descriptions.

**What (other) tasks could the dataset be used for?**

As discussed in the main paper, our multipurpose dataset can be used to research various learning tasks. These tasks include, but are not limited to:

- standard image classification,
- learning beyond hard labels [69],
- learning from noisy annotations [50],
- multi-annotator learning [68],
- annotation aggregation [73],
- and active learning [16].

**Is there anything about the composition of the dataset or the way it was collected and preprocessed/cleaned/labeled that might impact future uses?** *For example, is there anything that a dataset consumer might need to know to avoid uses that could result in unfair treatment of individuals or groups (e.g., stereotyping, quality of service issues) or other risks or harms (e.g., legal risks, financial harms)? If so, please provide a description. Is there anything a dataset consumer could do to mitigate these risks or harms?*

The observations and associated images used as task data were carefully curated from iNaturalist under their respective licenses. The licensing information is provided by iNaturalist users and cannot be verified with absolute certainty. If any potential license or copyright violations (cf. Appendix C)

are discovered, we will correct them promptly. In such a case, dataset consumers will be informed via GitHub and Zenodo to adjust their data accordingly.

**Are there tasks for which the dataset should not be used?** *If so, please provide a description.*

The dataset must be used exclusively for non-commercial research tasks. Moreover, the annotator metadata has been carefully designed and is not intended to motivate the inclusion of sensitive and protected personal data in similar collections.

**Any other comments?**

Dataset consumers must ensure their dataset's usage complies with all licenses, laws, regulations, and ethical guidelines (cf. Appendix C).

### B.6 Distribution

**Will the dataset be distributed to third parties outside of the entity (e.g., company, institution, organization) on behalf of which the dataset was created?** *If so, please provide a description.*

The dataset is accessible to everyone.

**How will the dataset be distributed (e.g., tarball on website, API, GitHub)?** *Does the dataset have a digital object identifier (DOI)?*

The dataset is available through Zenodo with a respective DOI. Code to load and experiment with the dataset is provided via GitHub.

**When will the dataset be distributed?**

The dataset is already publicly available.

**Will the dataset be distributed under a copyright or other intellectual property (IP) license, and/or under applicable terms of use (ToU)?** *If so, please describe this license and/or ToU, and provide a link or other access point to, or otherwise reproduce, any relevant licensing terms or ToU, as well as any fees associated with these restrictions.*

[4 (b, c)] Appendix C and Table 11 provide an overview of our dataset's licenses.

**Have any third parties imposed IP-based or other restrictions on the data associated with the instances?** *If so, please describe these restrictions, and provide a link or other access point to, or otherwise reproduce, any relevant licensing terms, as well as any fees associated with these restrictions.*

[4 (b, c)] Appendix C and Table 11 provide an overview of our dataset's licenses.

**Do any export controls or other regulatory restrictions apply to the dataset or to individual instances?** *If so, please describe these restrictions, and provide a link or other access point to, or otherwise reproduce, any supporting documentation.*

No.

**Any other comments?**

None.

### B.7 Maintenance

**How can the owner/curator/manager of the dataset be contacted (e.g., email address)?**

If questions or issues are relevant to other potential dataset consumers, we ask to create a corresponding issue at our GitHub repository (codebase). In all other cases, the dataset consumers can contact the dataset collectors via the e-mail `marek.herde@uni-kassel.de`.

**Is there an erratum?** *If so, please provide a link or other access point.*

No errors have been found so far.

**Will the dataset be updated (e.g., to correct labeling errors, add new instances, delete instances)?** *If so, please describe how often, by whom, and how updates will be communicated to dataset consumers (e.g., mailing list, GitHub)?*

Currently, no extensions are planned to ensure the comparability of future evaluation results on our dataset. However, if any errors are discovered, they will be corrected, and the dataset will be updated with a new version number on Zenodo. The change will also be announced in our GitHub repository.

**If the dataset relates to people, are there applicable limits on the retention of the data associated with the instances (e.g., were the individuals in question told that their data would be retained for a fixed period of time and then deleted)?** *If so, please describe these limits and explain how they will be enforced.*

In compliance with the GDPR (cf. Appendix D), all annotators were asked for their consent to use their contributions for scientific purposes. Additionally, all collected metadata from the annotators has been fully anonymized, so there are no restrictions on data retention.

**Will older versions of the dataset continue to be supported/hosted/maintained?** *If so, please describe how. If not, please describe how its obsolescence will be communicated to dataset consumers.*

All dataset versions will be available on Zenodo.

**If others want to extend/augment/build on/contribute to the dataset, is there a mechanism for them to do so?** *If so, please provide a description. Will these contributions be validated/verified? If so, please describe how. If not, why not? Is there a process for communicating/distributing these contributions to dataset consumers? If so, please provide a description.*

We warmly welcome scientific contributions from the community in the context of our dataset. Contributors are encouraged to create an issue on GitHub to discuss potential implementations. Code contributions can also be integrated via pull requests. For correctness and transparency, we and the community will publicly discuss and verify each contribution.

**Any other comments?**

None.

## C    Assets and Licenses

[4 (a, b)] This work integrates various well-established assets, i.e., platforms, code, and models, to facilitate the creation of our dataset `dopanim` and the associated empirical evaluation. In this way, we aim to ensure our results' correctness, including improved reproducibility and access to other users.

For the **collection and publication** of the task data, annotation data, and annotator metadata, we mostly relied on the following assets:

- *iNaturalist* [21] (platform, individual license) was the source for collecting the task data, e.g., images and their ground truth class labels.
- *pyinaturalist* [33] (code, MIT license) implements a Pythonic interface to the iNaturalist API and was used for downloading and filtering observations.
- *LabelStudio* [55] (platform, Apache-2.0 license) is an open-source tool for annotating various types of data. Its community edition served as the annotation platform for our work.
- *Zenodo* [5] (platform, individual license) is an open platform for sharing and preserving research outputs of various formats, including our dataset `dopanim`.

For the **empirical evaluation** consisting of a benchmark and case studies on seven variants of our dataset `dopanim`, we mostly relied on the following assets:

- *PyTorch* [35] (code, individual license) and *PyTorch Lightning* [11] (code, Apache-2.0 license) are central Python packages for implementing and training deep learning models, corresponding to multi-annotator learning approaches in our case.
- *DINOv2* [34] (model, Apache-2.0 license) offers multiple self-supervised learning models, of which we employed the ViT-S/14 model.

- *Hydra* [67] (code, MIT license) is a framework to configure complex applications corresponding to our work's experiments with their hyperparameters.
- *MLFlow* [71] (code, Apache-2.0 license) supports the organization of machine learning projects and assisted us with the logging and reading of the empirical results.
- *Scikit-learn* [36] (code, BSD-3-Clause license) is a famous machine learning framework, of which we used pre-processing techniques and logistic regression.
- *Scikit-activeml* [27] (code, BSD-3-Clause license) is a library of active learning algorithms employed by us for the use case on annotation times in active learning.

Further assets, such as NumPy [15] and Pandas [63] as standard Python packages in scientific computing, are included as requirements of working with our codebase.

[4 (b, c)] Table 11 provides an overview of our `dopanim` **dataset's** licenses. The task data retains the licenses from iNaturalist, while the annotation data and annotator metadata are collected by us and distributed under the license CC-BY-NC 4.0. Our **codebase** on GitHub to load and experiment with the dataset is available under the BSD-3-Clause license. As authors, we bear full responsibility for removing parts of our dataset or codebase or withdrawing our paper if confirmed violations of licensing agreements, intellectual property rights, or privacy rights are identified. In such a case, dataset consumers will be informed via GitHub and Zenodo to adjust their data accordingly. Moreover, dataset consumers are responsible for ensuring their dataset's usage complies with all licenses, laws, regulations, and ethical guidelines. In the case of any violation, we make no representations or warranties and accept no responsibility.

Table 11: License overview [4 (b, c)] – The left column lists the licenses, while the four right columns list the number of records for the respective data type. The task data is split into observation and image data because an observation and its associated image may have different licenses on iNaturalist [21].

| License | Observation Data Records [#] | Image Data Records [#] | Annotation Data Records [#] | Annotator Metadata Records [#] |
|---|---|---|---|---|
| CC0 | 601 | 347 | 0 | 0 |
| CC-BY | 1,241 | 1,257 | 0 | 0 |
| CC-BY-SA | 4 | 10 | 0 | 0 |
| CC-BY-NC | 13,837 | 14,057 | 52,795 | 20 |
| CC-BY-NC-SA | 51 | 63 | 0 | 0 |

# D   Broader Impact

Our annotation campaign for the `dopanim` dataset involved 20 human annotators providing over 52,000 annotations. This dataset offers valuable insights for advancements in machine learning research and raises important considerations in a broader context.

[1 (c, d)] Leveraging cost-efficient yet potentially error-prone annotators is vital in times of big data. However, requesting the support of human annotators, particularly through crowdsourcing, poses societal risks. On the one hand, human annotators can introduce noise into the dataset due to various reasons, e.g., missing expertise or lack of concentration [16]. Although our dataset is designed to mirror such human annotation noise, the reliance on such annotations requires careful reflection of their impact on generalization performance. On the other hand, annotation platforms often engage vulnerable individuals under challenging working conditions with low compensation [46]. It is essential to ensure fair compensation and working conditions. Despite its potential benefits, collecting metadata about annotators poses privacy risks. Including only relevant information and rigorously protecting the annotators' privacy is essential to prevent misuse and data breaches [65].

To mitigate the risks inherent to human annotators' work, we implemented the following policies:

- [5 (c)] Each annotator was employed as a student assistant at the University of Kassel and thus received a **fair compensation** following university standards. Specifically, student assistants without a first university degree received € 12.41 per hour, and those with a first university degree € 13.33 per hour. In both cases, the regulations of the German minimum wage were met.
- [4 (d, e)] We implemented **data privacy** by following the General Data Protection Regulation (GDPR) [41], which gave the annotators the right to reject the annotation work at any time

and ensures that no annotator can be identified after the publication of the dataset. For this purpose, each annotator received a GDPR-compliant document to explain all annotators' rights and the purpose of the annotation campaign. After voluntary consent, the personal data of the annotators was only stored on storage media within the University of Kassel for the organization of the annotation campaign. The collected annotator metadata does not contain this personal data; it only includes task-related data. In addition, all personal data that would have revealed a mapping between annotators and their true identities was deleted from the storage media before the dataset's publication.

- [5 (a)] Healthy **working conditions** were ensured on the one hand by German labor laws and on the other hand by additional freedoms granted as part of the annotation campaign. Considering these labor laws, the annotators were allowed to work remotely and freely organize their working hours. This implies, in particular, that breaks could be taken at any time. Moreover, the annotators had the right to withdraw from the annotation campaign.

These policies ensured ethical standards and integrity throughout our annotation campaign.

# E    Related Datasets: Addendum

Section 2 mainly describes related datasets that provide ground truth class labels alongside noisy annotations from multiple humans, are publicly available, and are frequently used in the literature for evaluating multi-annotator learning approaches. An exception is the `animal10n` dataset, included in Table 1 due to its similarity to the classification task of `dopanim`. The publicly available version of `animal10n` does not include information about which annotator provided which class label and is, thus, not used for evaluating multi-annotator learning approaches. Additional animal-related image datasets are `inat` [57] and `animal-web` [25], which provide only true class labels. Schmarje et al. [47] published a new collection of previously existing image datasets by adding annotations from multiple annotators. This collection is a benchmark to investigate the trade-off between the benefit of multiple annotations per instance and associated annotation costs (no annotation times). In contrast to `dopanim`, the datasets only contain hard class labels, and probabilistic class labels are obtained by combining class labels from multiple annotators. Two further datasets with class labels from multiple annotators are `compendium` and `mozilla` [19], where five annotators classified texts of software defect reports. Corresponding ground truth labels are unknown. None of these datasets provides annotator metadata.

# F    Empirical Evaluation: Addendum

This appendix details the main paper's experiments and associated results. More concretely, we discuss computational resources, present the hyperparameter search, and introduce annotator profiles.

## F.1    Computational Resources

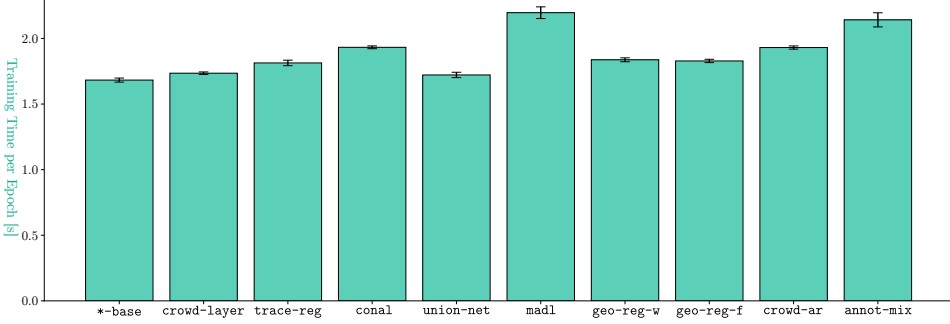

Figure 14: Training times [3 (d)] – The bars show the mean and standard deviation of the training time per epoch [s] for each approach across the seven `dopanim` variants. The times were recorded following the experimental setup of the main paper employing an AMD Ryzen 9 7950X as CPU.

[3 (d)] We primarily performed experiments on the Slurm cluster of the IES group at the University of Kassel. This cluster is equipped with multiple NVIDIA Tesla V100 GPU and NVIDIA Tesla A100 GPU servers for machine learning and, in particular, deep learning research. However, we needed such a GPU server only once to cache the images of our dataset `dopanim` as self-supervised features learned by the DINOv2 ViT-S/14 [34]. Subsequent benchmark and use case experiments were performed on CPU servers equipped with an AMD EPYC 7742 CPU. Additionally, we conducted small-scale experiments, e.g., creating the $t$-SNE [56] plot in the main paper, on a workstation equipped with an NVIDIA RTX 4090 GPU and an AMD Ryzen 9 7950X CPU. Using the CPU of this workstation, we measured the training times per epoch of the different approaches for the seven data set variants of `dopanim`. Figure 14 presents the corresponding results as a bar plot. We observe that all approaches have training times in a similar range, where the baselines are the fastest and `madl` the slowest. For an upper estimate of the computation time of all experiments, we use $3s$ as training time per epoch to account for any overhead. For 50 training epochs, $45 \cdot 10$ hyperparameter experiments, $7 \cdot 12 \cdot 10$ benchmark experiments, and $7 \cdot 5 \cdot 10$ use case experiments (ignoring the fast experiments on annotation times in active learning), we obtain a total computation time of about $68h$.

## F.2 Hyperparameter Search

[3 (b)] We specified learning rate, batch size, and weight decay as hyperparameters for the RAdam optimizer [31] in a grid search, where the validation accuracy (with early stopping) of the upper baseline `gt-base` was our objective. Table 12 presents the corresponding results. This procedure ensures that hyperparameters are found with which the classification task can be learned. In addition, none of the approaches is disadvantaged or favored. Furthermore, an individual hyperparameter study per approach is difficult in practical scenarios due to the lack of a validation set with ground truth class labels [70].

Table 12: Hyperparameter search for the upper baseline `gt-base` [3 (b)] – The column headings indicate the names of the tested hyperparameters, which are the learning rate (`lr`), training batch size (`bs`), and weight decay (`wd`). All results report the mean and standard deviation across ten repeated experiments. The **best** result across all hyperparameter configurations is marked.

| lr | (128, 0) | (128, .0001) | (128, .00001) | (32, 0) | (32, .0001) | (32, .00001) | (64, 0) | (64, .0001) | (64, .00001) |
|---|---|---|---|---|---|---|---|---|---|
| | (bs, wd) | | | | | | | | |
| | Validation Accuracy (acc [%]) ↑ | | | | | | | | |
| .001 | $89.5_{\pm0.2}$ | $89.4_{\pm0.3}$ | $89.4_{\pm0.2}$ | $89.2_{\pm0.3}$ | $89.2_{\pm0.5}$ | $89.2_{\pm0.4}$ | $\mathbf{89.5_{\pm0.4}}$ | $89.3_{\pm0.4}$ | $89.4_{\pm0.4}$ |
| .005 | $89.1_{\pm0.3}$ | $89.1_{\pm0.2}$ | $89.1_{\pm0.4}$ | $89.1_{\pm0.5}$ | $89.2_{\pm0.4}$ | $89.2_{\pm0.5}$ | $89.0_{\pm0.4}$ | $89.0_{\pm0.4}$ | $89.2_{\pm0.3}$ |
| .01 | $88.9_{\pm0.4}$ | $89.2_{\pm0.6}$ | $89.1_{\pm0.3}$ | $89.3_{\pm0.4}$ | $89.3_{\pm0.4}$ | $89.2_{\pm0.5}$ | $89.0_{\pm0.4}$ | $89.1_{\pm0.4}$ | $89.1_{\pm0.3}$ |
| .05 | $88.7_{\pm0.5}$ | $88.9_{\pm0.4}$ | $88.8_{\pm0.3}$ | $88.5_{\pm0.4}$ | $88.0_{\pm0.6}$ | $88.5_{\pm0.4}$ | $88.7_{\pm0.3}$ | $89.2_{\pm0.3}$ | $88.6_{\pm0.5}$ |
| .1 | $88.9_{\pm0.3}$ | $88.6_{\pm0.3}$ | $88.7_{\pm0.5}$ | $86.2_{\pm0.5}$ | $85.4_{\pm0.7}$ | $86.6_{\pm0.8}$ | $88.4_{\pm0.5}$ | $88.0_{\pm0.6}$ | $88.3_{\pm0.4}$ |

## F.3 Annotator Profiles

In the main paper, we presented the confusion matrix of the top-label predictions, the reliability diagram of the human-estimated likelihoods, and the histogram of annotation times across all annotators. Since the annotation behaviors between the annotators are quite different, we now present these three aspects individually for each annotator. Together with selected metadata, we thus obtain a kind of profile per annotator in Figs. 15, 16, 17, and 18.

The analysis of the **annotation times** reveals that most annotators process the majority of their annotation tasks in rather short time frames ($5s$ to $10s$). However, a few annotators, e.g., `sunlit-sorcerer` (cf. Fig. 15) and `echo-eclipse` (cf. Fig. 16), take considerably more time to annotate an image. In combination with annotation times as a critical cost factor, active learning strategies [16] could aim to prefer fast-working annotators.

The variation between the individual **reliability diagrams** [10] of the annotators is large. On the one hand, there are some annotators for which higher top-label confidences imply higher accuracies, e.g., `galactic-gardener` and `cosmic-wanderer` (cf. Fig. 18). On the other hand, there are also annotators whose top-label confidences are very unreliable, e.g., `dreamy-drifter` (cf. Fig. 17) and `twilight-traveler` (cf. Fig. 18). Quantifying the reliabilities and biases of such human-estimated likelihoods poses challenging research problems for the future.

We observe differences in error patterns by comparing the annotators' individual **confusion matrices**. For example, the annotator `sunlit-sorcerer` (cf. Fig. 15) is an expert on distinguishing insects but struggles with correctly annotating squirrels. In contrast, the annotator `sapphire-sphinx` (cf. Fig. 16) confuses insects more frequently. Such observations underline the importance of modeling annotator performance at least as class-dependent.

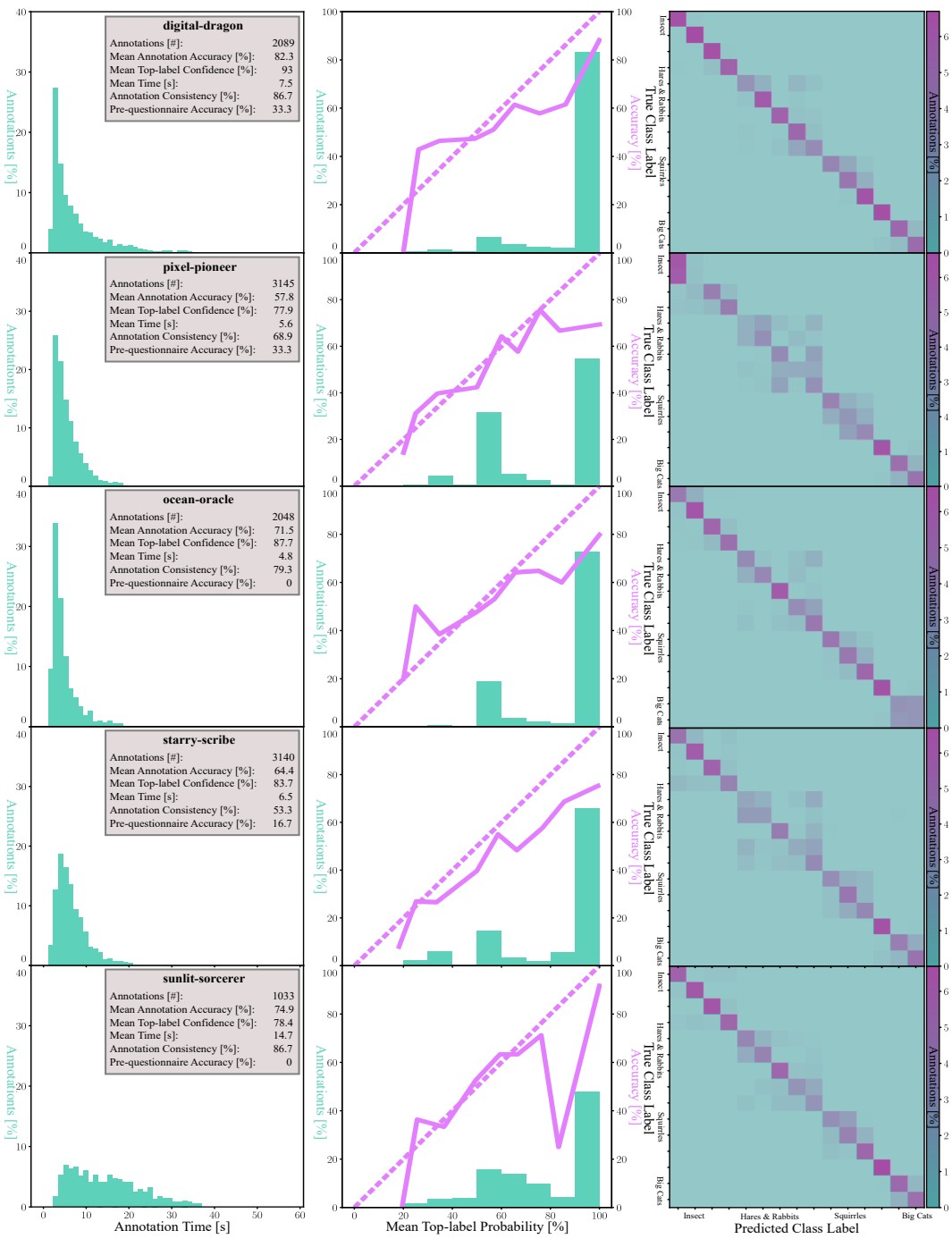

Figure 15: Annotator profiles I – Each annotator is profiled by their individual histogram of annotation times, reliability diagram of likelihoods, and confusion matrix. These graphs are supplemented by metadata computed from annotation data or extracted from completed questionnaires.

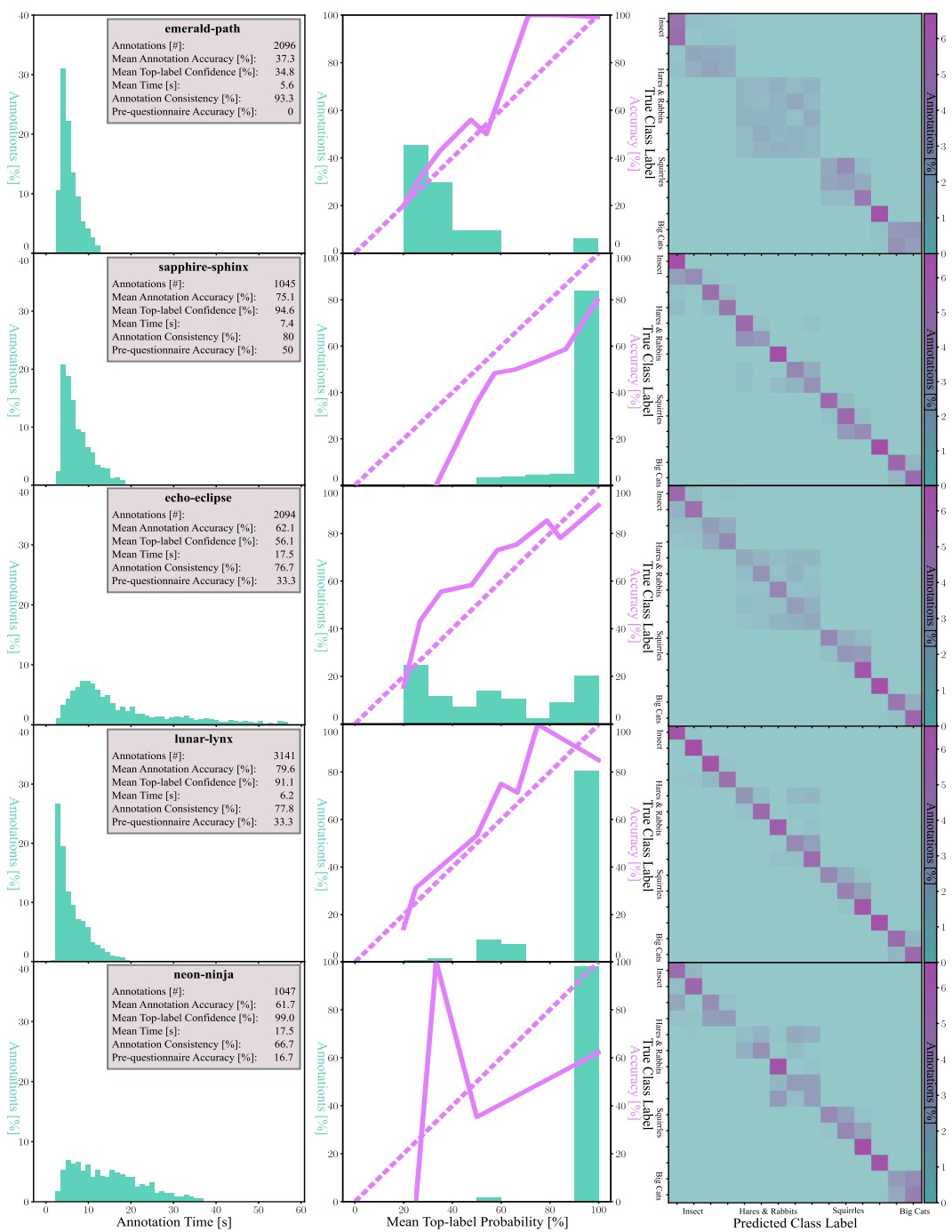

Figure 16: Annotator profiles II – Each annotator is profiled by their individual histogram of annotation times, reliability diagram of likelihoods, and confusion matrix. These graphs are supplemented by metadata computed from annotation data or extracted from completed questionnaires.

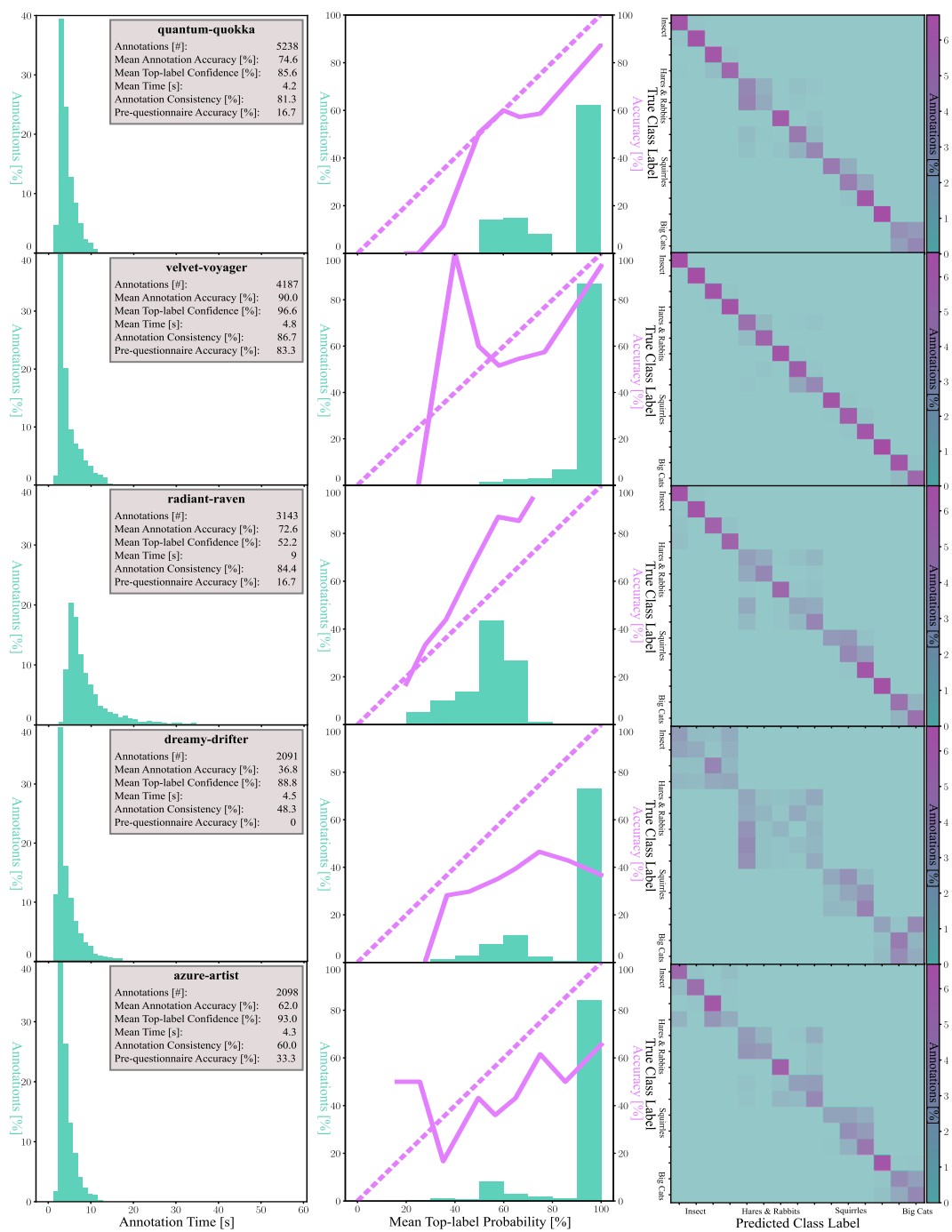

Figure 17: Annotator profiles III – Each annotator is profiled by their individual histogram of annotation times, reliability diagram of likelihoods, and confusion matrix. These graphs are supplemented by metadata computed from annotation data or extracted from completed questionnaires.

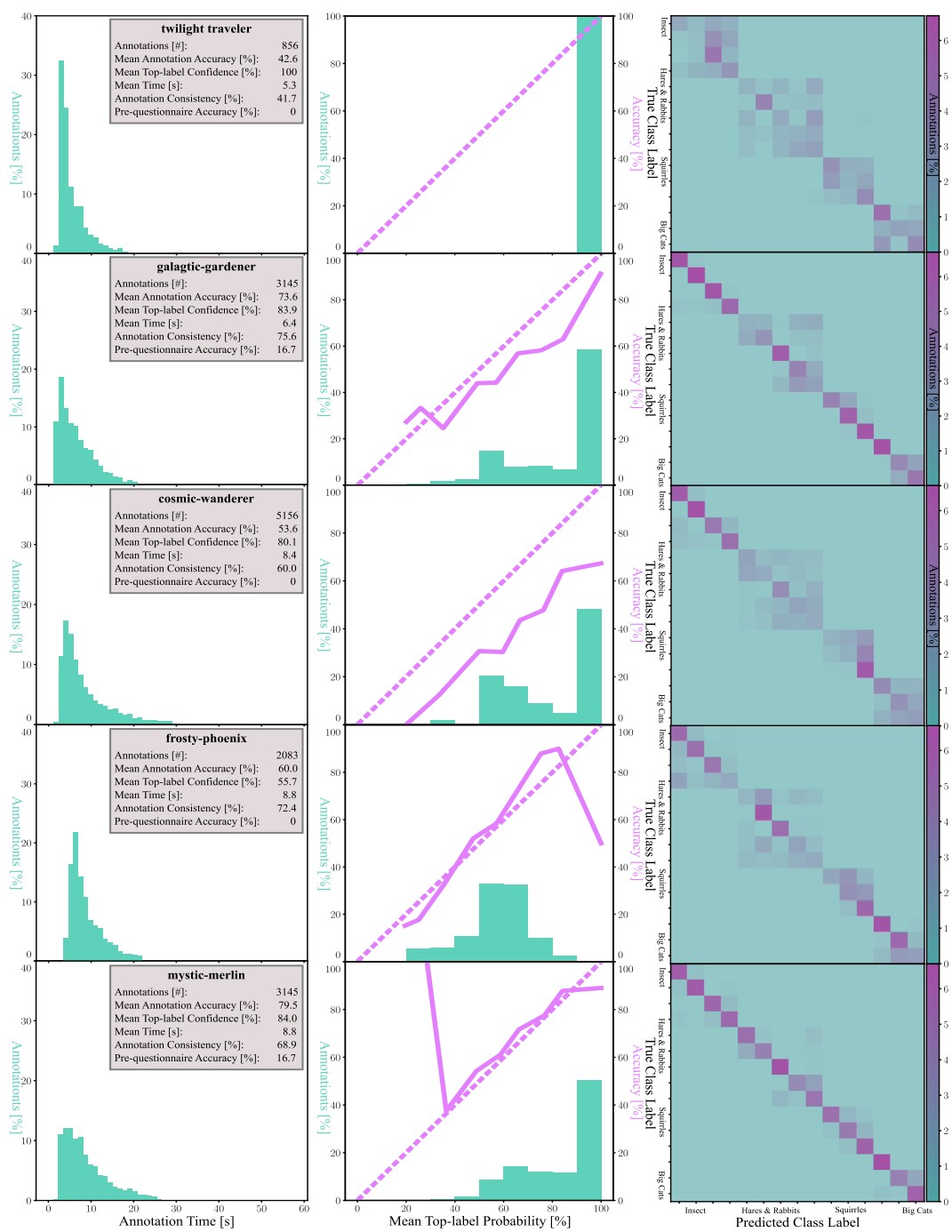

Figure 18: Annotator profiles IV – Each annotator is profiled by their individual histogram of annotation times, reliability diagram of likelihoods, and confusion matrix. These graphs are supplemented by metadata computed from annotation data or extracted from completed questionnaires.

