# OpenReview forum: "dopanim: A Dataset of Doppelganger Animals with Noisy Annotations from Multiple Humans"
_NeurIPS.cc/2024/Datasets_and_Benchmarks_Track — NeurIPS 2024 Track Datasets and Benchmarks Poster_

### Official Review · Reviewer_4EnK · 2024-07-01
**Review dpanim**

**Rating:** 7
**Confidence:** 4
**Correctness:** Okay
**Clarity:** Okay

**Review:**

Overall, I really like the idea of multiple annotations per annotator and think the work is well written and well structured. My main issue is about the difference to previous works. They show in table 1 and in the appendix many very similar datasets and the main question is dopanim unique enough with respect to previous work. I think it is worth accepting but needs to work on this distinction more to improve. Details about strength and recommendations of changes below.

**Strengths:**

- The idea is great and especially the idea of doppelganger animals are very refreshing
- The paper has a good outline and especially the grey boxes for key insights and feature improve the reading massivly.
- I like that you give transparency about related work of your own group.
- The methodolgy is sound and the evalualtions seem mainly very good (except for the example given below).

**Additional Feedback:**

N/A

**Documentation:**

Looks fine

**Ethics:**

They take explizitly licenced images from iNaturalist.

**Limitations:**

Limitations are well described

**Opportunities For Improvement:**

My main issue is the question is this work different enough from previous work. Here are some comments which I think you can do to improve your work:
- Table 1 and the additional works in the addendum seem to arbitary spllit. Personally, I found the comparisons in the addendum more interesting than the 4 variants of cifar10/cifar100. The works in the addendum have a large variety and seem also be more closely related with regard to your domain (e.g. animal10n). Please integrate them in the table
- You claim that no other dataset has annotator meta-data. I agree that these results are not publicly available but based on quick glance at the annotator setup I would be pretty sure that the authors of [18] and [7] are aware of the characteristics of the annotators since they have only a few individual annotators like yourself. Have you reached out to verify this? If this is true it would degrade your claim of difference slightly. I still see that you are the ones who actually released the information but it would good to clarify if this data is in theory available.
- In 5.1 you state individual human uncertainty is important, Have you any comparison to averaged uncertainty like in CIFARH? What about mentioned papers where individual uncertainty is calculated via averaging across multiple annotations per annotator. How do these setups compare? I think you never mention the annotation cost difference between adjusting several sliders and just giving a hard label. I would assume that your approach is better than averaging across all annotators or multiple annotations per annotator but what is the exact trade off with regard to annotation cost and gained performance. If this trade-off is not favorable (e.g. you gain 1% accuracy but need to increase the cost by factor 10) then it would mean that one of your claimed major benefits of soft individual uncertainty is not a major benefit at all.

Other recommendations for improvement:
- Please explain shortly the methods form table 4. Without a short explanation a comparison with regard to similarities and differences is not easy doable except for reading all referenced papers.
- Have you a baseline with your averaged soft labels instead of majority vote aggregration? It seems like a more natural baseline like majority vote which throughs all individual information away.
- Increase the size of all tables and figures in line 290. They are too small for good readability.
- Please explain more how the meta-data is used in section 5.2. I can certaintly read the other paper of your working group and puzzle it out but the paper should be written in a self-contained way.

**Relation To Prior Work:**

See opportunities for improvement

**Summary And Contributions:**

The authors present a new dataset about doppelganger animals with multiple annotations per image. The main contributions over other similar datasets are soft per person uncertainties, annotation times and annotator meta-data. The evaluation shows multiple interesting research questions which can be potentially answered with this data.

---

> ### Author Rebuttal · Authors · 2024-08-17
>
> Dear Reviewer,
>
> Thanks for your thorough review. We're happy that you like the general idea and presentation of our paper. Below, we describe how we plan to implement your suggested _opportunities for improvement_ by numbering our answers with (1)-(7) to match your 7 bullet points. Depending on the final decision, we plan to incorporate any proposed changes, including additional content like more detailed explanations, in the camera-ready version, utilizing the extra page if needed.
>
> **(1) Related Work:**
>
> _Explanations_: Table 1 includes the most commonly used datasets in multi-annotator learning [1,2], which contain ground truth labels and are easily accessible without additional permissions. Due to space constraints, we moved other related datasets to the supplementary material. However, we agree with you and recognize the importance of task domain similarities (animal classification).
>
> _Changes_: We plan to add relevant datasets, like animal10n, to Table 1. To do so, we'll follow reviewer ogMH26's suggestion, and also add a row indicating the availability of ground truth labels. The description of less related datasets will remain in the supplementary material, which will be indicated in the main paper for interested readers.
>
> **(2) Annotator Metadata:**
>
> _Explanation_: We downloaded and inspected the data together with the papers' information but could not identify any task-related annotator metadata.
>
> _Changes_: Following your suggestion, we asked the corresponding authors of [3] and [4] regarding any potential unpublished annotator metadata.  The corresponding author of [3] replied that they did not consider such an idea when publishing the data and, therefore, have only the very general but unpublished information whether an annotator is a trained expert or not. In contrast, our metadata is particularly extensive and task-specific. This was only possible because we implemented special questionnaires before and after the annotation tasks. In case we receive a reply from the corresponding author of [4] during the discussion phase, we'll post an update comment.
>
> **(3 + 5) Probabilistic Labels:**
>
> _Explanations_: You are right to question the trade-off between the benefit of querying probabilistic class labels (likelihoods) and their annotation costs compared to hard labels. Yet, our study's primary goal was to determine whether there is a potential benefit in querying human-estimated probabilities. Moreover, we have queried probabilities for all classes to allow simulating other (less informative) label types, e.g., partial labels or top label confidences.
>
> _Changes_: We'll emphasize our purpose of collecting likelihoods across all potential classes more strongly in Section 3.2 and point out the potential higher annotation costs, which is subject to future research, as a limitation in Section 6. Further, we have attached a PDF showing increasing annotation times (in seconds) with an increasing entropy of the likelihoods. Yet, such a plot does not directly quantify the actual amount of higher costs for obtaining likelihoods because also requesting hard class labels can take more time for uncertain images. Finally, we evaluated your suggestion of comparing *soft majority voting* [5] when using the hard vs. probabilistic labels (normalized likelihoods). The table below reports the test accuracies in [%], confirming the benefit of probabilistic labels in term of generalization performance. We agree with you that this a more suitable comparison for the case study in Section 5.1 such that we will adjust this subsection accordingly. Furthermore, we think that soft majority voting with hard class labels is also an important baseline for Section 4, where we'll also add the obtained results.
>
> | _Dataset Variant_ | _Hard Labels_  | _Probabilistic Labels_ |
> |---------------------|------------------|--------------------------|
> | worst-1             | $27.0 \pm 0.6$ | $39.2 \pm 0.7$         |
> | worst-2             | $45.2 \pm 0.4$ | $55.9 \pm 0.3$         |
> | worst-v             | $57.3 \pm 0.4$ | $65.7 \pm 0.4$         |
> | rand-1              | $74.3 \pm 0.4$ | $76.1 \pm 0.2$         |
> | rand-2              | $77.9 \pm 0.6$ | $78.7 \pm 0.2$         |
> | rand-v              | $79.2 \pm 0.4$ | $80.3 \pm 0.3$         |
> | full                | $81.1 \pm 0.3$ | $81.7 \pm 0.3$         |
>
>
> **(4 + 7) Multi-annotor Learning Techniques**:
>
> _Explanations_: Space constraints were the main reason for not going into more detail when introducing the foundations of multi-annotator learning techniques in Section 4.1.
>
> _Changes_: We plan to add a table overviewing the main differences between the techniques' properties, e.g., type of annotator performance, training procedure, parameter initialization, loss function, and regularization terms. In this context, it will also be clearer that some of these architectures allow training with multi-modal data as input, consisting of the instance data (e.g., images or text) and annotator data (e.g., task-specific annotator metadata in the form of feature vectors).
>
>
> **(6) Fontsizes**:
>
> _Changes_: Thanks for the hint. We'll increase the fontisze such that one can read the text without zooming.
>
> **References**
> - [1] Z. Cao et al. Learning from Crowds with Annotation Reliability. In SIGIR Conf. Res. Dev. Inf. Retr., 2023.
> - [2] F. Rodrigues et al. Learning from multiple annotators: Distinguishing good from random labelers. Pattern Recognit. Lett., 34(12): 1428-1436, 2013.
> - [3] L. Schmarje et al. Is one annotation enough? A data-centric image classification benchmark for noisy and ambiguous label estimation. In NeurIPS, 2022.
> - [4] J. Hernández-González et al. Two datasets of defect reports labeled by a crowd of annotators of unknown reliability. Data Brief, 18:840–845, 2018.

---

> > ### Comment · Reviewer_4EnK · 2024-08-19
> > **increased score**
> >
> > Thank you for your detailed Answers. I see you put a lot of work and improvements into the reply this I increased my score. I personally still see room for improvement with regard to probalistic data and related work structure but I acknowledge you made well informed and valuable decisions this I increased my score nevertheless.
> >
> > Good luck with the submission!

---

### Official Review · Reviewer_ana5 · 2024-07-21
**A good work, but some details should be clarified.**

**Rating:** 7
**Confidence:** 3
**Correctness:** Yes
**Clarity:** Yes

**Review:**

Please check the comments below.

**Strengths:**

- The collection of multi-annotator annotations is valuable since it reflects human perceptions of fine-grained categories.
- The paper is well-written, and the data-sheet is clear.

**Additional Feedback:**

Please check the Opportunities For Improvement.

**Documentation:**

Yes

**Limitations:**

Yes

**Opportunities For Improvement:**

This study is interesting and has a good quality. However, some details should be clarified.
## Weaknesses
- The reviewer suggests the authors clarify the major differences between this study and previous work [1].
[1] Is one annotation enough? - A data-centric image classification benchmark for noisy and ambiguous label estimation, 2022 NeurIPS
- It would be better to provide a comparison with other animal classification datasets, such as AnimalWeb [2] and ANIMAL-10N [3].
[2] AnimalWeb: A Large-Scale Hierarchical Dataset of Annotated Animal Faces, 2020 CVPR
[3] Selfie: Refurbishing unclean samples for robust deep learning, 2019 ICML
- Why are the standard deviations of annotations per annotator and accuracy per annotator quite large?
- How should the annotator disagreements for the test set be handled? Does the accuracy well reflect the performance of models under noisy annotations?
- The comparison between the annotations provided in this study and the annotations provided in iNaturalist is missing.

**Relation To Prior Work:**

Please check the Opportunities For Improvement.

**Summary And Contributions:**

This study introduces a doppelganger animal dataset with multiple annotations per image, aiming to advance research in machine learning with noisy annotations from multiple annotators.

---

> ### Author Rebuttal · Authors · 2024-08-17
>
> Dear Reviewer,
>
> Thanks for taking the time to review our paper. We're pleased that you acknowledge the valuable effort of collecting annotations from multiple annotators. Below, we refer with (1)-(5) to your 5 bullet points under weaknesses and detail how we aim to address them. Depending on the final decision, we plan to incorporate any proposed changes, including additional content like more detailed explanations, in the camera-ready version, utilizing the extra page if needed.
>
> **(1 + 2) Related Work:**
>
> _Explanations_: In our related work, we mainly focused on the datasets used in other multi-annotator learning works due to space constraints. Differences to further related works, e.g., [1] and [2], are highlighted in the related work addendum (Section E in the supplementary material). Yet, your questions are reasonable and aligns with other reviewers. Therefore, we briefly summarize the main differences to your mentioned papers:
>
> - The published datasets in [1,2,3] do not contain any annotator metada with task-related background information and provide only hard class labels from the individual annotators, while we provide annotator metadata and likelihoods estimated by individual annotators.
> - The datasets in  [2, 3] do not provide the information which annotation stems from which annotator, which would be crucial for estimating annotator performances in multi-annotator learning. Although, the datasets in [1] contain this information, there are no multi-annotator learning approaches evaluated.
> - The paper [3] does not provide ground truth class labels from real experts such that noise levels can only be estimated.
> - The paper [2] does not consider any noisy label setting.
>
> _Changes_: We'll expand our discussion of related works by moving parts of the related work addendum from the supplementary material to our main paper.
>
> **(3) Standard Deviations:**
>
> _Explanations_: There are multiple reasons why these standard deviations are quite large:
>
> - The numbers of annotations vary, since our annotators received payments on an hourly basis and further agreed on individual numbers of working hours. Accordingly, the number of annotations fit these working hours. However, scenarios where there is an imbalance in the number of annotations between annotators is a common issue in annotation campaigns [4]. Accordingly, our dataset allows evaluating whether techniques can handle such scenarios.
>
> - The accuracies across annotators vary because the annotators had different levels of motivation, concentration, expertise, etc. Particularly, the expertise is an important factor in our task of classifying doppelganger animals because it requires specialized knowledge about the individual animal's characteristics. We equipped the annotators with such knowledge by providing corresponding tutorials (cf. lines 114-117). However, not all annotators received the same tutorials, e.g., some got specialized tutorials on big cats or squirrels. This way, we ensured different levels of expertise among the annotators, which is a common observation in annotation campaigns [5].
>
> _Changes_: We'll briefly mention these two aspects as part of the annotation data analysis in Section 2.2.
>
> **(4) Disagreement in Test Set:**
>
> _Explanations_: The validation and test set contain only the labels, which were obtained from the iNaturalist platform. We refer to these labels as ground truth labels because we collected only images with research-grade labels (cf. lines 102-104) ensuring the labels' correctness. The very high quality of these research-grade labels was also studied in research, e.g., the authors of [6] found that 98.3% of their sample set with research-grade labels were correct. As a result, we can assume that performance measurements for the validation and test set accurately reflect the model's actual generalization performances.
>
> _Changes_: We will add a brief sentence in combination with reference [6] such that potential readers are aware of the very high quality of the validation and test labels in Section 3.1.
>
> **(5) Comparison to iNaturalist**:
>
> _Explanations_: The iNaturalist species classification and detection dataset [7] targets image classification with large class imbalance. For this purpose, the datasets comprises many images from thousands of animal classes only with their research-grade label as reliable approximation of the ground truth [6]. Accordingly, this dataset is not suited to study learning with noisy class labels.
>
> _Changes_: We will make this distinction more clear in Section 2, when briefly talking about other animal classification datasets.
>
> **References:**
> - [1] L. Schmarje et al. Is one annotation enough? A data-centric image classification benchmark for noisy and ambiguous label estimation. In NeurIPS, 2022.
> - [2] M. H. Khan et al. AnimalWeb: A Large-Scale Hierarchical Dataset of Annotated Animal Faces. In CVPR, 2020.
> - [3] H. Song et al. Selfie: Refurbishing Unclean Samples for Robust Deep Learning. In ICML, 2019.
> - [4] K. Wazny et al. "Crowdsourcing" ten years in: A review. J. Glob. Health 7(2): 2017.
> - [5] F. Rodrigues et al. Learning from multiple annotators: Distinguishing good from random labelers. Pattern Recognit. Lett., 34(12): 1428-1436, 2013.
> - [6] A. Garretson et al. Citizen science data reveal regional heterogeneity in phenological response to climate in the large milkweed bug, Oncopeltus fasciatus. Ecol. Evol. 13(7): 2023.
> - [7] Grant Van Horn et al. The INaturalist Species Classification and Detection Dataset. In Conf. Comput. Vis. Pattern Recognit., pp. 8769-8778, 2018.

---

> > ### Comment · Reviewer_ana5 · 2024-08-19
> > **Response to rebuttal by authors**
> >
> > The authors have addressed the concerns. The reviewer will maintain the rating as 7. Thanks.

---

### Official Review · Reviewer_F9JK · 2024-07-22
**Interesting approach with reasonable decisions, though the motivation could be clearer.**

**Rating:** 6
**Confidence:** 3
**Correctness:** I did not find any obvious issues abo…

**Review:**

The authors set out to tackle two key problems: label noise in the annotation process and improvements in multi-annotator learning. To address these, they created Dopanim by relying on annotators' likelihood information instead of the conventional binary decisions. They also collected additional annotator-dependent information to further demonstrate the utility of their approach.

I started reading the paper with great excitement because the impact of the annotation process on model quality is an understudied area. While the authors put significant effort into demonstrating the utility of their approach and made several decisions in data collection and experimental methodologies, the motivation behind these decisions was not adequately presented. I will expand upon this point in the 'Opportunities for Improvement' section.

**Strengths:**

- Deep learning models have a strong capacity to learn hidden assumptions embedded in the modeling process. Therefore, it is essential to revisit the assumptions behind dataset creation, especially those related to annotations. This paper is a valuable contribution in this regard.

- The authors made an extra effort to highlight their critical takeaway messages, which aided my understanding of the extensive information provided in the paper.

- Despite concerns about scalability, their exploration of using human likelihood estimates is interesting, as datasets with low data and class counts could benefit from this approach.

- The authors made a conscientious effort to demonstrate the utility of their dataset curation process.

**Additional Feedback:**

N/A

**Clarity:**

While the authors put in a lot of effort by highlighting key takeaway messages, the breadth of the paper affected the presentation. Since I liked some of the ideas, I did spend some time on the paper and I believe that its presentation could be improved. I have highlighted some points that can help in my earlier responses.

**Documentation:**

I paper and accompanying GitHub repo do provide good information for other researchers to build on this work.

**Ethics:**

I do not find any ethical issues based on the responses in the checklist. Also, the dataset was derived from a public dataset.

**Limitations:**

The authors did thoughtfully discuss some key limitations of their work.

**Opportunities For Improvement:**

- Although the paper covers multiple research problems, as acknowledged by the authors, this breadth makes it difficult to read and affects the motivation and support for some decisions in dataset curation and experiments. The takeaway messages help, but a narrower, deeper focus would be beneficial. For example, concentrating on multi-annotator learning issues and their approach could better introduce Section 4's techniques and the reasoning behind the seven dataset variants.

- The paper narrates the experiments and key takeaways but lacks adequate motivation for the decisions. For instance, why were 15 animals and four categories chosen? Does this imply the research is only beneficial for fine-grained annotation tasks? The annotation interface sliders suggest an upper limit to likelihoods—how was this determined? I assume the ground truth from the iNaturalist dataset was used. If this is the case, there is an implicit assumption that the iNaturalist ground truth has negligible label noise. Is it conceivable that the annotation process for iNaturalist also suffered from the doppelganger effect to some extent?

- The confusion matrix in Figure 4 reflects the authors' conclusions about sources of confusion, but the result seems expected and possibly an artifact of the experimental design. Why not use an existing dataset without creating a new one with 15 classes?

- I have concerns about the utility of annotation time. While it's been used in other papers, it assumes annotators aren't distracted by other factors, making it noisy despite containing some information.

- While the authors demonstrate the benefits of their strategy for majority voting, they could explore the utility of 'selection frequency' as an additional benchmark. This approach, which measures what fraction of annotators selected a particular class for an image, has been explored in several studies[1,2,3,4] and could provide a robust comparison to demonstrate the advantages of their likelihood approach to dataset annotation.

- I reviewed a paper that collects full human label distributions for CIFAR-10. Though the paper is cited as the paper [25], the similarities and contrasts in the annotation process were not discussed. Given the differences in achieving full distributions, this work could serve as a strong candidate to demonstrate the advantages of the authors' approach.

Overall, I liked some of the ideas in the paper, but I struggled with their presentation and the paper's positioning.

[1] Do ImageNet Classifiers Generalize to ImageNet? Benjamin Recht, Rebecca Roelofs, Ludwig Schmidt, and Vaishaal Shankar
Proceedings of the 36th International Conference on Machine Learning, Long Beach, California, PMLR 97, 2019. Copyright 2019 by the author(s).

[2] From ImageNet to Image Classification: Contextualizing Progress on Benchmarks
Dimitris Tsipras, Shibani Santurkar, Logan Engstrom, Andrew Ilyas, Aleksander Madry Proceedings of the 37th International Conference on Machine Learning, PMLR 119:9625-9635, 2020.

[3] Identifying Statistical Bias in Dataset Replication
Logan Engstrom, Andrew Ilyas, Shibani Santurkar, Dimitris Tsipras, Jacob Steinhardt, Aleksander Madry Proceedings of the 37th International Conference on Machine Learning, PMLR 119:2922-2932, 2020.

[4] What Makes ImageNet Look Unlike LAION
Ali Shirali and Moritz Hardt, arXiv

**Relation To Prior Work:**

While the authors did provide some related works, a concern I had was the lack of comparison with related works that explored the 'selective frequency.'

**Summary And Contributions:**

The paper presents a dataset named Dopanim, containing 15,750 animal images divided into four groups and fifteen classes. Each group features visually similar animals. Instead of binary decisions, annotators provided likelihoods for all categories. Multiple annotators labeled each image, and their metadata was collected. The authors evaluated common multi-annotator learning methods and found that the likelihood annotation approach helps mitigate label noise. They highlighted use cases where Dopanim's annotation data is beneficial: using human likelihoods as weights in majority voting, enhancing learning with annotator metadata, and supporting active learning.

---

> ### Author Rebuttal · Authors · 2024-08-17
>
> Dear Reviewer,
>
> Thanks for your thoughtful review. We're pleased that you liked several ideas of our paper. Below, we explain how we address your 6 bullet points under _opportunities for improvement_, to which we refer by (1)-(6). Depending on the final decision, we plan to incorporate any proposed changes, including additional content like more detailed explanations, in the camera-ready version, utilizing the extra page if needed.
>
> **(1) Paper's Scope**:
>
> _Explanations_: We cover multiple research problems to demonstrate the broad applicability of our dataset (see contribution (3)). Due to space constraints, we sometimes provided only high-level explanations and condensed takeaways. We believe this broad scope helps readers determine how the dataset might benefit their research. However, we acknowledge that this approach may make some details in the multi-annotator learning benchmark in Section 4 harder to grasp.
>
> _Changes_: As a compromise, we'll detail the benchmark. Concretely, we'll add a table overviewing the main differences between the multi-annotator learning techniques properties, e.g., type of annotator performance, training procedure, loss function, and regularization terms. Despite these potential changes, we think that keeping our message of a multipurpose research dataset is beneficial for the research community.
>
> **(2) Decisions' Motivation**:
>
> We try to clarify your questions regarding our decisions in the following:
>
> - Most datasets with noisy annotators focus on simple tasks like generic object classification. We aimed to define a task requiring specialized knowledge, where errors stem from varying expertise levels (cf. lines 59-62). We selected four animal groups that are difficult to distinguish, limiting the task to 15 classes due to budget constraints, as more classes would require more images and thus higher annotation costs.
> - Our dataset can be used to study annotation problems with fewer classes, e.g., when training and testing only with the images of one of the four groups. However, could you detail what you exactly mean by fine-grained annotation?
> - The absolute values and the limit of 10 are rather irrelevant; only the relative comparison between likelihoods matters (see Fig. 3 caption), e.g., assigning 1 to both Leopard and Jaguar is identical to assigning 10 to the two animals. Both cases reflect maximum uncertainty between the two classes. Further examples were given in the tutorial to each annotator.
> - The images were sourced from the iNaturalist platform, not the dataset [1], due to the dataset's limited number of images per class (e.g., only 33 training images of a Jaguar), which complicates noisy label learning. We collected only images with research-grade labels, which have been shown to be very reliable, e.g., 98.3% correct in a sample set of [2].
>
> _Changes_: We'll incorporate these motivations for our decisions in condensed forms in Section 3. For example, we'll briefly add that other studies have confirmed the reliability of research-grade labels to Section 3.1.
>
> **(3) Confusion Matrix**:
>
>  _Explanations_: We tried to emphasize the usage of new data in (2: Decisions' Motivation). Of course, you are right that by design, the confusion matrix should reflect more annotation errors within a group of doppelganger animals. Yet, showing this matrix helps to confirm our intuitive understanding of the annotators' noise patterns.
>
>  _Changes_: In case of space issues when incorporating other changes, we'll move the confusion matrix to the supplementary material.
>
> **(4) Annotation Time**:
>
> _Explanations_: Measuring meaningful annotation times is challenging. Therefore, we provided each annotator with a detailed tutorial on using LabelStudio, including guidance on doing breaks. Annotators could ask questions to avoid misunderstandings. The results of the active learning experiments (see Section 5.3) support the quality of the annotation times, as uncertainty sampling consistently selected images with higher annotation times, which is unlikely to occur by chance.
>
> _Changes_: We'll add this explanation to our datasheet and a remark in the main paper in Section 5.3
>
> **(5) Selection Frequency:**
>
> _Explanations_: We like your suggestion as further comparison.
>
> _Changes_: We performed experiments using the hard labels to count votes, but also the normalized likelihoods (probabilistic labels) as soft votes. We used the `Threshold0.7` [3] approach, training only with images with a selection frequency of at least 0.7. The table below confirms the test accuracy gains [%] of using selection frequencies combined with probabilistic labels and will be added to our supplementary.
>
> |_Dataset Variant_|_Hard Labels_|_Probabilistic Labels_|
> |-|-|-|
> |worst-1|$27.0\pm0.6$|$37.5\pm0.5$|
> |worst-2|$49.3\pm0.4$|$57.0\pm0.3$|
> |worst-v|$57.0\pm0.4$|$64.2\pm0.6$|
> |rand-1|$74.3\pm0.4$|$76.2\pm0.3$|
> |rand-2|$79.2\pm0.3$|$79.1\pm0.2$|
> |rand-v|$78.8\pm0.4$|$80.2\pm0.3$|
> |full|$80.6\pm0.3$|$81.2\pm0.4$|
>
>
> **(6) Related Work:**
>
> _Explanations_: Next to the different classification task and missing annotator metadata, dopanim differs from cifar10h [4] by providing likelihoods from each individual annotator. In contrast, cifar10h contains only hard labels from each annotator and soft labels are only obtained by normalizing the votes across multiple annotators per image.
>
> _Changes_: We agree with you and highlight the difference between the label type of cifar10h and dopanim in Section 2.
>
> **References:**
>
> - [1] G. Van Horn et al. The INaturalist Species Classification and Detection Dataset. In CVPR, 2018.
> - [2] A. Garretson et al. Citizen science data reveal regional heterogeneity in phenological response to climate in the large milkweed bug, Oncopeltus fasciatus. Ecol. Evol. 13(7): 2023.
> - [3] B. Recht et al. "Do imagenet classifiers generalize to imagenet?." In ICML, 2019.
> - [4] J. C. Peterson et al. Human Uncertainty Makes Classification More Robust. In ICCV, 2019.

---

> > ### Comment · Reviewer_F9JK · 2024-08-20
> >
> > I appreciate the authors' thoughtful responses to the feedback provided and their proposed revisions. I apologize for the earlier confusion regarding the term "fine-grained annotation"; I had intended to refer to fine-grain image classification. Overall, I am satisfied with the authors' responses to my review comments, and I have decided to revise my score upward by one point.
> > The assumptions underlying dataset creation are critical, especially when working with deep learning models that have strong learning capabilities. In my review, I implicitly raised concerns about the applicability of this work to public datasets that were not curated in the specific manner used in this study. While I understand the need to constrain research experiments, I was curious why the authors did not apply their techniques to an existing vision dataset of a similar scale without restricting their work to a fine-grain image classification scenario.
> > Despite this concern, I believe the paper makes a valuable contribution to the field of fine-grained image recognition, and its clarity will be improved by the proposed revisions.

---

### Official Review · Reviewer_ogMH · 2024-07-26
**Excellent paper; its not only a dataset but also a very comprehensive benchmark, with multiple clear usecases and great documentation and reproducibility**

**Rating:** 9
**Confidence:** 4
**Correctness:** Yes. The paper seems extremely compre…

**Review:**

The dataset provided is very detailed, with datasheet, useable code, and complete: annotator statistics, and even clearly displayed what UI was used for annotation. It is very comprehensive.

**Strengths:**

- The paper showcases 1 major usecases and 3 smaller usescases, illustrating the usefulness of the collected data
- In the major usecase, 8 algorithms are benchmarked and 2 baselines are benchmarked. Thus the paper also provides a comprehensive becnhmark.
- The authors indicate clearly which algorithms are their own in name of transparancy.
- The paper contrasts the dataset with other available datasets
- The data collection is very comprehensive and complete.
- The github provides scripts to reproduce the results. Even the interfaces are shared that were used for LabelMe.

**Additional Feedback:**

A very nice work; the reproducibility of this dataset has one of the highest standards that I have seen.

**Clarity:**

Yes, very well written. Easy to read, very detailed, clearly stated research questions, and nice emphasis added via a box to highlight main contributions, etc.

**Documentation:**

Looks very comprehensive.

**Ethics:**

No.

**Limitations:**

not relevant

**Opportunities For Improvement:**

- One thing that I did not like is that the comparison with animal10n is only done in the Appendix, because the comparison in section 2 only pertains to dataset where groundtruth is known. However, this criteria is not mentioned in section 2 (only in the appendix) and from the main paper its not clear that animal10n exists (a citation to it is also missing in the main body). However, in my opinion, its pretty clear that this is the most similar dataset. I find it very misleading. I think this paper, and a citation to it, should definitely be added to the main body, and the dataset should be added to Table 1, with another extra criteria / row that indicates where the groundtruth is known or not.

- In the field of social signal processing, where machine learning is for example used to classify emotions, annotations also often disagree. This field may be of particular interest to the authors, as it can further motivate multi annotation learning. A related dataset where multi annotation data was collected is [1] (but maybe it was not written in the paper, but I know the authors have the data available).

[1] Vargas-Quiros, J., Cabrera-Quiros, L., Oertel, C., & Hung, H. (2023). Impact of annotation modality on label quality and model performance in the automatic assessment of laughter in-the-wild. IEEE Transactions on Affective Computing.

**Relation To Prior Work:**

I don't like that the discussion with the previous animal dataset is missing.

**Summary And Contributions:**

This paper provides a new dataset with multiple annotations. The main contribution is that all the different annotators' their annotations are saved, so that they can be used for research purposes. For example, annotation time, different labels, and even annotator data (confidence, experience) are saved. The paper demonstrates the usefulness of this data by demonstrating one major usecase: using multiple annotations for "multiple annotation learning", and 3 smaller usecases: improving a voting baseline using multiple labels, using annotator metadata to improve multi-annotator learning, and finally, seeing if "active learning" algorithms tend to select samples that take longer to annotate.

---

> ### Author Rebuttal · Authors · 2024-08-17
>
> Dear Reviewer,
>
> Thanks for your thorough review and your positive feedback. Below, we number our answers with (1)-(2) to address your 2 bullet points in _opportunities for improvement_. Depending on the final decision, we plan to incorporate any proposed changes, including additional content like more detailed explanations, in the camera-ready version, utilizing the extra page if needed.
>
> **(1) Related Work:**
>
> _Explanations_: Our original motivation was to create Table 1 with the most commonly used datasets in multi-annotator learning [1,2], which contain ground truth labels and are easily accessible without additional permissions. Due to space constraints, we moved other related datasets to the supplementary material. However, we totally agree with you and also other reviewers who mentioned this, that we should also consider the similarities in the task domain (animal classification).
>
> _Changes_: We aim to follow your suggestion by adding animal10n [3] to Table 1:
>
> |    **Dataset Overview**    |     $\texttt{animal10n}$    |
> |:----------------------------------|--------------------:|
> | data modality                   |      images      |
> | training instances [#]       |     $50,000$    |
> | validation instances [#]    |         ✗      |
> | test instances [#]             |     $5,000$    |
> | classes [#]                |        $10$       |
> | ground truth labels                |      ✗          |
> | annotators [#]                   |             $15$        |
> | platform                   |        ✗       |
> | annotator metadata         |         ✗        |
> | annotation times           |        ✗       |
> | soft class labels          |         ✗        |
> | annotations per instance [#]     | $1_{\pm 0.00}$ |
> | annotations per annotator [#]   |         ✗        |
> | overall accuracy [%]       |       $92$       |
> | accuracy per annotator [%] |         ✗        |
>
> Since the authors of animal10n [3] did not provide individual annotators' class labels, we asked the corresponding authors whether they have access to the information to fill in the remaining gaps in the above table, e.g., the used platform and the actual number of annotations per annotator (after their filtering step). The annotators' accuracies are unknown due to missing ground truth. We update the table once, we receive a reply. Beyond this table update, we'll add an explicit statement to Section 2 indicating that potential other related datasets are part of the supplementary material.
>
> **(2) Social Signal Processing:**
>
> _Explanations_: The paper [4] addresses indeed a very interesting topic in multi-annotator learning. In our opinion, dealing with multi-modal data representing different perspectives when annotating poses another challenge in this research area. Moreover, the annotation of laughter intensity is likely a more subjective task than recognizing nominal classes of animals.
>
> _Changes_: We would propose to include the paper [4] in Section 6, because multi-annotator learning with more subjective and particularly multi-modal data seems to be underexplored and, therefore, an interesting topic for future work. Since we plan to continue the development of our codebase on multi-annotator learning and datasets with annotations from multiple annotators, we've already checked the GitHub repository of the paper [4], where indeed annotation results seem to be available as .csv files. However, due to the more complex setting of multi-modal data and the limited time, we could not yet work in deeper.
>
> **References**
> - [1] Z. Cao et al. Learning from Crowds with Annotation Reliability. In SIGIR Conf. Res. Dev. Inf. Retr., 2023.
> - [2] F. Rodrigues et al. Learning from multiple annotators: Distinguishing good from random labelers. Pattern Recognit. Lett., 34(12): 1428-1436, 2013.
> - [3] H. Song et al. Selfie: Refurbishing Unclean Samples for Robust Deep Learning. In ICML., 2019.
> - [4] J. Vargas-Quiros et al. Impact of annotation modality on label quality and model performance in the automatic assessment of laughter in-the-wild. IEEE Trans. Affect. Comput., 15(2): 519-534, 2024.

---

### Author Rebuttal · Authors · 2024-08-17

Dear Reviewers,

Thanks again for all your insightful reviews helping us to improve the paper and for acknowledging several strengths, e.g., that the "collection of multi-annotator annotations is valuable" [ana5], our "data collection is very comprehensive" [ogMH], the "exploration of using human likelihoods is interesting" [F9JK], and "the evaluations seem mainly very good" [4EnK]. Although, we have already prepared detailed individual rebuttals to address your specific concerns, this general response aims to briefly outline the main takeaways regarding your proposed opportunities for improvement.

**Opportunities for Improvement:**

Please note that any changes adding extra content to the main paper will be compensated by the inclusion of an additional page (if accepted). Additionally, we could move Fig. 2 and/or Fig. 4 to the supplementary material if necessary.

- _Related Work_ [ogMH, F9JK, ana5, 4EnK]: We'll expand Section 2 ("Related Datasets") in the main paper by incorporating elements from Section E of the supplementary material. Specifically, we will emphasize the differences between our dataset, dopanim, and other animal datasets [1,2,3], such as by including animal10n in Table 1 (cf. rebuttal for reviewer ogMH), and highlight distinctions from datasets that measure uncertainties by averaging hard class labels from multiple annotators [4,5]. Pointing out these differences will also improve understanding the motivation behind our decisions when describing the data collection in Section 3.
- _Benchmark_ [F9JK, 4EnK]: We'll expand our explanations in Section 4 ("Benchmark: Multi-annotator Learning"), specifically highlighting the key differences in the properties of the evaluated multi-annotator learning techniques, such as annotator performance types, training procedures, parameter initialization, loss functions, and regularization terms. This will help readers better understand the empirical results.
- _Probabilistic Labels_ [F9JK, 4EnK]: Valuable suggestions for additional comparisons were made, such as soft majority voting (including averaging probabilistic labels) and selection frequency [6] for filtering training data. We have already evaluated these suggestions and included the corresponding results in tables within the individual rebuttals. The accuracy gains from using the likelihoods of individual annotators have been confirmed. These results will be discussed and partially included in Section 5.1. Due to space limitations, some of these new results will be moved to the supplementary material with a reference note in the main paper.

**Summary**:

By incorporating the outlined changes, we are confident that our paper's quality will be enhanced while preserving its core contributions, particularly introducing a dataset suited for various research problems (Section 5) with a focus on benchmarking multi-annotator learning techniques (Section 4).

**Discussion**:

We look forward to receiving your feedback on our individual rebuttals and are happy to answer any follow-up questions/concerns you may have.

**References**:
- [1] H. Song et al. Selfie: Refurbishing Unclean Samples for Robust Deep Learning. In ICML, 2019.
- [2] Grant Van Horn et al. The INaturalist Species Classification and Detection Dataset. In Conf. Comput. Vis. Pattern Recognit., pp. 8769-8778, 2018.
- [3] M. H. Khan et al. AnimalWeb: A Large-Scale Hierarchical Dataset of Annotated Animal Faces. In CVPR, 2020.
- [4] L. Schmarje et al. Is one annotation enough? A data-centric image classification benchmark for noisy and ambiguous label estimation. In NeurIPS, 2022.
- [5] J. C. Peterson et al. Human Uncertainty Makes Classification More Robust. In ICCV, 2019.
- [6] B. Recht et al. "Do imagenet classifiers generalize to imagenet?." In ICML, 2019.

---

> ### Author Response · Authors · 2024-09-01
> **Rebuttal Recap**
>
> Dear Reviewers,
>
> We **thank** you for your effort to provide feedback regarding our rebuttal.
>
> Meanwhile, the **opportunities for improvements** (identified by you and summarized in our general rebuttal) have
> been incorporated into our paper and its supplementary material. For example, along with the updated results on probabilistic
> labels (cf. responses to Reviewer 4EnK and F9JK) and the expanded dataset table (cf. response to Reviewer ogMH), we have
> extended the benchmark by (a slightly different $\LaTeX$ version of) the following table:
>
> | _Approach_             | _Venue_    | _Annotator Performance Assumption_ | _Annotator Performance Modelling_                    | _Loss Function_                 | _Can process annotator metadata?_ |
> |------------------------|------------|------------------------------------|------------------------------------------------------|---------------------------------|----------------------|
> | $\texttt{cl}$          | AAAI 2018  | class-dependent                    | noise adaption layer per annotator                   | cross-entropy                   | ✗                    |
> | $\texttt{trace-reg}$   | CVPR 2019  | class-dependent                    | confusion matrix per annotator                       | cross-entropy + regularization  | ✗                    |
> | $\texttt{conal}$       | AAAI 2021  | class-dependent                    | noise adaption layers per and across annotators | cross-entropy + regularization  | ✔                    |
> | $\texttt{union-net}$   | TNNLS 2022 | class-dependent                    | noise adaption layer across annotators               | cross-entropy                   | ✗                    |
> | $\texttt{madl}$        | TMLR 2023  | instance-dependent                 | confusion matrix per instance-annotator pair         | cross-entropy + regularization  | ✔                    |
> | $\texttt{geo-reg-w/f}$ | ICLR 2023  | class-dependent                    | confusion matrix per annotator                       | cross-entropy + regularization  | ✗                    |
> | $\texttt{crowd-ar}$    | SIGIR 2023 | instance-dependent                 | reliability scalar per instance-annotator pair       | cross-entropy sum of two models | ✔                    |
> | $\texttt{annot-mix}$   | ECAI 2024  | instance-dependent                 | confusion matrix per instance-annotator pair         | cross-entropy                   | ✔                    |
>
> This table highlights the differences in how various one-stage multi-annotator learning approaches model annotator
> performance, whether annotator metadata can be considered, and which loss functions are used.
>
> **Edit:** _A checkmark (✔) in the annotator metadata column indicates that the approach can process such data in principle. However, in the paper where the respective approach was presented, no real dataset with real annotator metadata was used._
>
> We have added more detailed explanations, e.g., on the type of multi-annotator specific regularization, in the paper and the supplementary material.
>
> Finally, there were two further aspects in our rebuttal for which we had promised an update:
> - The statistics for the $\texttt{animal10n}$ dataset [1] were still incomplete in our response to Reviewer ogMH. We
> have now received confirmation that the authors will send us the annotator-specific labels to complete these statistics as soon as they locate them.
> - Following the suggestion of Reviewer 4EnK, we asked the authors of [2] about any unpublished annotator metadata,
> but unfortunately, none data that can be linked to individual annotators are available.
>
> In **summary**, the revisions we have made have improved our paper's quality without altering its core contributions
> acknowledged by your reviews.
>
> **References:**
> - [1] H. Song et al. Selfie: Refurbishing Unclean Samples for Robust Deep Learning. In ICML., 2019.
> - [2] J. Hernández-González et al. Two datasets of defect reports labeled by a crowd of annotators of unknown
>   reliability. Data Brief, 18:840–845, 2018.

---

### Decision · Program_Chairs · 2024-09-26

**Decision:**

Accept (Poster)

**Comment:**

This paper received unanimous final acceptance votes from the reviewers, who reached a consensus that reflects it as a solid submission. The reviewers appreciated the paper for addressing interesting topics in fine-grained image classification, the comprehensive process with the authors' great efforts, and the sound evaluations' correctness. This AC agrees with these views and further commends the elegantly designed data collection process, which serves multiple purposes with specified use cases. The corresponding analysis offers valuable insights for future dataset construction.

Additionally, the presentation is very clear, providing detailed guidance on using the dataset, supported by a well-presented evaluation guide and the provided GitHub repository. The authors also effectively addressed all reviewer concerns, with most reviewers expressing satisfaction with the responses. While some reviewers raised concerns about the scalability of the work, which this AC acknowledges, this issue can be beneficially addressed in future work; the paper's strengths outweigh the minor weaknesses. I believe the reviewers provided reasonable suggestions to improve the work, which makes it more solid. The authors are encouraged to consider the improvements, as they have already addressed some of these points in the supplementary materials.

As a result, this AC strongly champions this submission for acceptance. It is a strong work overall.